# Insights into fungal diversity of a shallow-water hydrothermal vent field at Kueishan Island, Taiwan by culture-based and metabarcoding analyses

Ka-Lai Pang[1]*, Sheng-Yu Guo[1], I-An Chen[1], Gäetan Burgaud[2], Zhu-Hua Luo[3], Hans U. Dahms[4], Jiang-Shiou Hwang[1], Yi-Li Lin[1], Jian-Shun Huang[1], Tsz-Wai Ho[5], Ling-Ming Tsang[6], Michael Wai-Lun Chiang[7], Hyo-Jung Cha[1]

1 Institute of Marine Biology and Centre of Excellence of the Oceans, National Taiwan Ocean University, Keelung, Taiwan, 2 Laboratoire Universitaire de Biodiversité et Ecologie Microbienne, Université de Brest, Plouzané, France, 3 Key Laboratory of Marine Biogenetic Resources, Third Institute of Oceanography, Ministry of Natural Resources, Xiamen, China, 4 Department of Biomedical Science and Environment Biology, College of Life Science, Kaohsiung Medical University, Kaohsiung, Taiwan, 5 School of Biological Sciences, University of Western Australia, Perth, Australia, 6 School of Biological Sciences, Chinese University of Hong Kong, Kowloon Tong, Hong Kong SAR, 7 Department of Chemistry, City University of Hong Kong, Kowloon Tong, Hong Kong SAR

* klpang@ntou.edu.tw

**Data Availability Statement:** All relevant data are within the paper and its Supporting Information file. The generated ITS sequences were published

## Abstract

This paper reports the diversity of fungi associated with substrates collected at a shallow hydrothermal vent field at Kueishan Island, Taiwan, using both culture-based and metabarcoding methods. Culture of fungi from yellow sediment (with visible sulfur granules), black sediment (no visible sulfur granules), the vent crab *Xenograpsus testudinatus*, seawater and, animal egg samples resulted in a total of 94 isolates. Species identification based on the internal transcribed spacer regions of the rDNA revealed that the yellow sediment samples had the highest species richness with 25 species, followed by the black sediment (23) and the crab (13). The Ascomycota was dominant over the Basidiomycota; the dominant orders were Agaricales, Capnodiales, Eurotiales, Hypocreales, Pleosporales, Polyporales and Xylariales. *Hortaea werneckii* was the only common fungus isolated from the crab, seawater, yellow and black sediment samples. The metabarcoding analysis amplifying a small fragment of the rDNA (from 18S to 5.8S) recovered 7–27 species from the black sediment and 12–27 species from the yellow sediment samples and all species belonged to the Ascomycota and the Basidiomycota. In the yellow sediments, the dominant order was Pleosporales and this order was also dominant in the black sediment together with Sporidiobolales. Based on the results from both methods, 54 and 49 species were found in the black and yellow sediments, respectively. Overall, a higher proportion of Ascomycota (~70%) over Basidiomycota was recovered in the yellow sediment and the two phyla were equally abundant in the black sediment. The top five dominant fungal orders in descending order based on species richness were Pleosporales>Eurotiales>Polyporales>Hypocreales>Capnodiales in the black sediment samples, and Polyporales>Pleosporales>Eurotiales>Capnodiales>Hypocreales in the yellow sediment samples. This study is the first to observe a high diversity

at NCBI GenBank under the accession numbers listed in the S1 Table.

**Funding:** Funding: Ka-Lai Pang thanks the Ministry of Science of Technology, Taiwan for financial support (MOST105-2621-M-019-002-, MOST106-2621-M-019-002-).

**Competing interests:** Competing interests: The authors have declared that no competing interests exist.

of fungi associated with various substrates at a marine shallow water hydrothermal vent ecosystem. While some fungi found in this study were terrestrial species and their airborne spores might have been deposited into the marine sediment, several pathogenic fungi of animals, including *Acremonium* spp., *Aspergillus* spp., *Fusarium* spp., *Malassezia* spp., *Hortaea werneckii*, *Parengyodontium album*, and *Westerdykella dispersa*, were recovered suggesting that these fungi may be able to cause diseases of marine animals.

## Introduction

Numerous studies have highlighted diverse marine fungi with important ecological roles such as commensals or pathogens of marine animals including corals [1] and sponges [2,3], trophic linkers between phytoplankton and zooplankton [4], or even nutrient recyclers. A relatively small percentage of described fungal species appears to be associated with the marine environment with just 1255 species [5]. In terms of biomass, marine fungi (together with marine protists) have been recently estimated to represent ~3% of the ~550 Gt carbon on Earth [6]. However, these estimates may be greatly underestimated due to under-sampling of diverse marine habitats. Yet, marine fungi are not represented in ocean ecosystem models [7], despite growing evidence of diverse marine fungi in the ocean.

The deep-sea (loosely defined as habitats below the epipelagic zone) represents the largest biome on Earth, representing more than 65% of the Earth's surface and more than 95% of the global biosphere [8]. The deep-sea encompasses a huge variety of ecological niches characterized by site-specific physical and geochemical conditions [9]. Deep-sea microorganisms, depending on the habitat, thus face many challenges such as elevated hydrostatic pressures, extreme temperature gradients and variable sea salt concentrations [10]. The deep subseafloor together with different deep-sea hotspots of life such as deep-sea hydrothermal vents, deep-hypersaline anoxic basins or even cold seeps have recently been reported to host active fungal communities with different ecological roles including recycling of refractory organic matter or competition with prokaryotes [11–20]. Burgaud and Edgcomb [21] have recently summarized all studies dedicated to the analysis of fungal communities from deep-sea and deep subsurface habitats (including both sedimentary and oceanic crustal habitats) and highlighted that culture-based approaches used so far have allowed the isolation of >116 species, a large majority of them being ubiquitous and affiliated to 11 fungal classes, with Eurotiomycetes, Dothideomycetes, and Microbotryomycetes being the most abundant. A "core culturable fungal diversity" consisting of 14 filamentous fungi (*Acremonium* sp., *Aspergillus glaucus* (also known as *Eurotium herbariorum*), *As. restrictus*, *As. sydowii*, *Aureobasidium pullulans*, *Cladosporium cladosporioides*, *Cl. sphaerospermum*, *Cordyceps confragosa* (also known as *Lecanicillium lecanii*), *Cyphellophora europaea*, *Exophiala dermatitidis*, *Exophiala* sp., *Penicillium chrysogenum*, *Pe. citrinum* and *Purpureocillium lilacinum*) and 3 yeasts (*Candida parapsilosis*, *Meyerozyma guilliermondii* and *Rhodotorula mucilaginosa*) was also established. Contrasted results were obtained using molecular studies. Indeed, based on 12 different studies specifically targeting deep-sea fungi, the fungal diversity was shown to be dominated by the classes Sordariomycetes, Dothideomycetes, Eurotiomycetes, Agaricomycetes, Saccharomycetes and Leotiomycetes, with *Aspergillus*, *Cryptococcus*, *Penicillium*, *Rhodotorula*, *Candida*, *Trichosporon*, *Cladosporium*, *Phoma*, *Exophiala*, *Fusarium* and *Malassezia* as the 10 most represented genera. Some studies have also revealed basal fungal lineages in the deep-sea, such as the Chytridiomycota [22], and the Cryptomycota [23]. While species richness appears higher in deep-sea

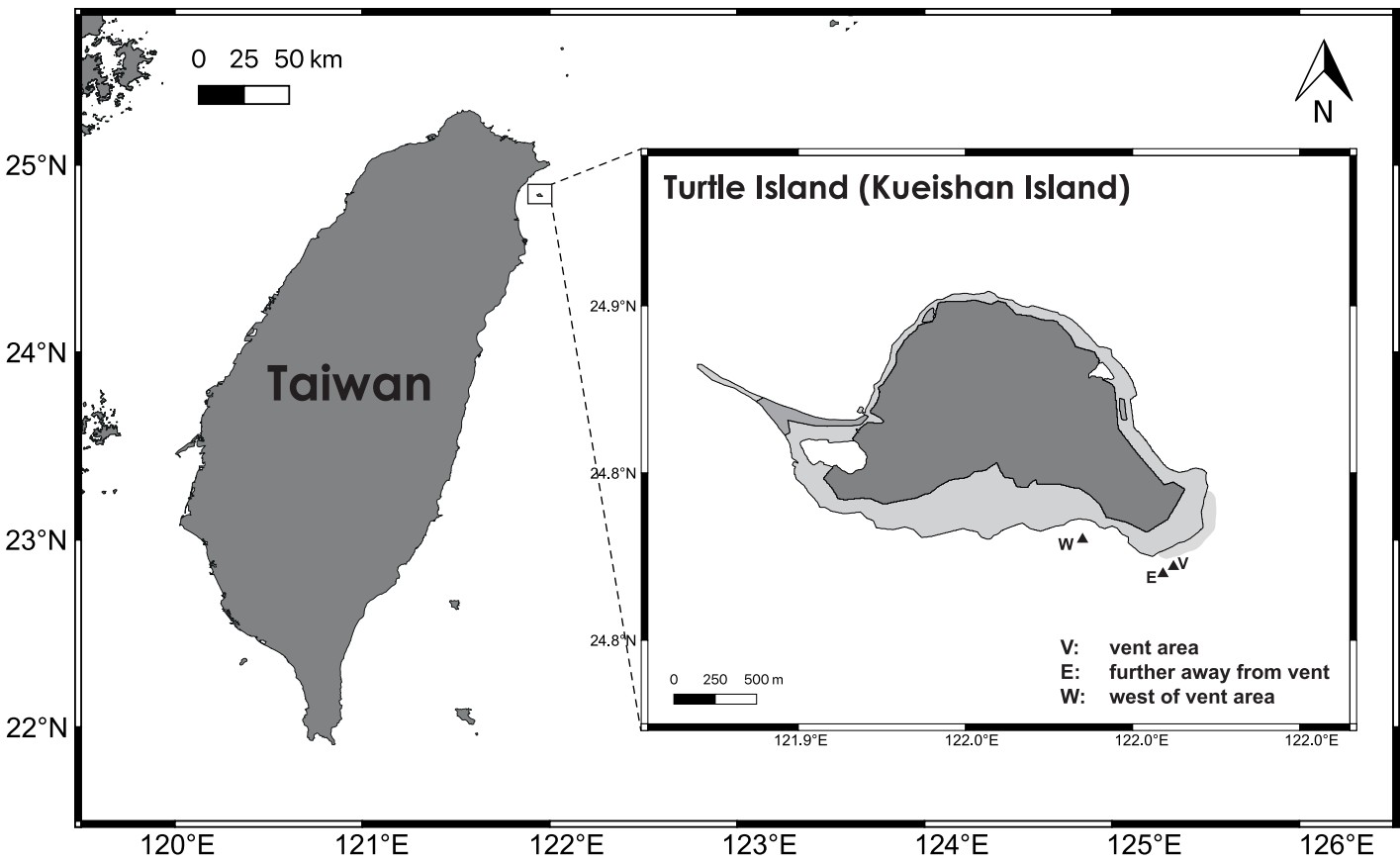

**Fig 1. Shallow hydrothermal vent field at Kueishan Island, Yilan County, Taiwan.** Sample location: V = vent area, E = further away from vent, W = west of vent area.

sediments compared to deep-sea vents, the uniqueness of these samples in terms of taxonomic composition appears higher in deep-sea vent samples, especially taking into account the recent description of new deep-sea hydrothermal vent species, for example *Candida oceani* [24] and *Yamadazyma barbieri* [25]. While deep-sea hydrothermal vents appear to be unique ecological niches for fungi, no in-depth investigation of fungal communities occurring in shallow hydrothermal vents has been processed so far, to the best of our knowledge.

Kueishan Island, also known as Turtle Island, is a volcanic island lying just outside Yilan County, Taiwan (Fig 1). In the southern end of the island, there are roughly 50 hydrothermal vent systems, with depth ranging from 10 m to 80 m, constantly emitting hydrothermal fluids (fluid temperature between 48 °C and 116 °C) and volcanic gases composed of high levels of carbon dioxide and hydrogen sulfide [26,27]. Many macro-organisms have been reported near this shallow-water hydrothermal vent system including fishes, crabs, mussels, sea anemones, snails, sipunculid worms, algae and zooplankton [26, 28,29]. However, knowledge on marine microorganisms including bacteria, archaea, and fungi is deficient. Recently, Wang et al. [30], using the pyrosequencing of the 16S rRNA gene, found that the Epsilonproteobacteria such as *Sulfurovum* and *Sulfurimonas* dominated the bacterial community in sediment samples collected near the hydrothermal vents at Kueishan Island. Also, they detected the presence of chemoautotrophic carbon fixation genes by the Epsilonproteobacteria in the samples suggesting their possible participation in the reductive tricarboxylic acid and the Calvin-Benson-Bassham cycles in the primary production in this extreme habitat. Complex microbial communities are

supposed to occur here with either chemoautotrophic and photoautotrophic primary producers along with heterotrophs, such as fungi. Recently, Jiang et al. [31] reported the isolation of *Aspergillus* spp. from sediment samples and the hydrothermal vent crab *Xenograpsus testudinatus* at Kueishan Island, which can be seen as first hints of fungal presence in this habitat. Here, we report the first comprehensive study of the diversity of fungi on various substrates at Kueishan Island. The main objective was to investigate the diversity of fungi in sediment, seawater and organic matter collected near the hydrothermal vents using culture-based and metabarcoding approaches.

## Materials and methods

### Collection of samples

The location of the shallow hydrothermal vents at Kueishan Island, Yilan, Taiwan is shown in Fig 1. Samples were collected on five collection trips: 23/6/2015 (yellow (Location V, Fig 1) and black (Location W, Fig 1) sediment), 10/10/2015 (yellow sediment, seawater, the crab *Xenograpsus testudinatus* (Location V, Fig 1)), 22-23/08/2016 (yellow and black sediment, water, crabs (Location E, Fig 1)), 29-30/03/2017 (black sediment, water, crabs, animal eggs (Location V, Fig 1)) and 28/06/2017 (yellow sediment, crabs (Location V, Fig 1)). Visible sulfur granules were found in the yellow sediment samples (with a higher $SO_3$ content) but not in the black sediment samples (with a lower $SO_3$ content) [30]. The sediment, seawater, crab (collected by hands of divers) and animal egg samples were immediately transferred into 50-ml sterile universal bottles. No permit was required to collect the animal samples as the collection site is not a national park. The crab *X. testudinatus* is not an endangered/protected species in Taiwan. The samples were kept in an ice bucket and brought to the laboratory for fungal isolation and molecular analysis. For each of the twenty-two sediment subsamples (in universal bottles) collected on the five collection dates, half of the sediment was transferred aseptically to another sterile universal bottle and freeze-dried for DNA extraction while the other half was used for isolation. A summary of the samples used in the isolation of fungi and the metabarcoding analysis (those with positive results) is shown in Table 1.

### Isolation

Different methods were used to isolate fungi from the seawater samples collected on different dates; undiluted/diluted or concentrated (by centrifugation and resuspension in a small volume of seawater) seawater was directly inoculated into one-fifth strength GYPS (0.8 g/L glucose, 0.8 g/L yeast extract, 0.4 g/L peptone, 1 L natural seawater)/CDS (Czapek-Dox prepared

**Table 1. Samples used in the culture-based and metabarcoding analyses.** Sample code: Y = yellow sediment sample, B = black sediment sample, V = vent area, E = further away from vent, W = west of vent area, 15 = collected in 2015, 16 = collected in 2016, 17 = collected in 2017.

| Date of sampling | Culture-based | | | | | Metabarcoding (Sample name) | |
|---|---|---|---|---|---|---|---|
| | Yellow sediment | Black sediment | Crab | Animal egg | Seawater | Yellow sediment | Black sediment |
| 23/6/2015 | | | | | | × (Y-V-15) | × (B-W-15) |
| 10/10/2015 | × (V) | | × (V) | | × (V) | | |
| 22/08/2016 | | × (E) | | | | | × (B-E-16) |
| 23/08/2016 | × (V) | | | | | | |
| 29/03/2017 | | × (V) | | | × (V) | | × (B-V-17) |
| 30/03/2017 | | | × (V) | × (V) | | | |
| 28/06/2017 | × (V) | | × (V) | | | × (Y-V-17) | |

with natural seawater instead of distilled water) liquid/solid media supplemented with 0.5 g/L each of Penicillin G sodium salt and streptomycin sulfate.

For the sediment samples, 1 g of sediment was directly inoculated into 50 mL GYPS and CDS liquid media supplemented with the antibiotics. The sediment samples were also serially diluted to $1 \times 10^{-3}$ with 100 µL of each dilution plated on GYPS and CDS agar media supplemented with the antibiotics. For some of the sediment samples with a liquid portion (seawater), the liquid (100 µL) was directly plated on GYPS and CDS agar media with the antibiotics.

The crab samples (all dead at the time of isolation) were crushed in sterile seawater with 1 g inoculated into 50 mL GYPS and CDS liquid media with the antibiotics. The crushed crab samples were serially diluted up to $1 \times 10^{-3}$ with 100 µL of each dilution plated on GYPS and CDS seawater agar media supplemented with the antibiotics. The crabs were also cut in halves with each half inoculated into GYPS and CDS liquid media with the antibiotics. The animal egg samples (collected as one clump) were directly inoculated into 50 mL GYPS and CDS liquid media with the antibiotics.

All inoculated samples were incubated at 25 ˚C for up to two weeks. Colonies appearing on the agar media were subcultured onto fresh GYPS agar plates as pure cultures. For the liquid media, 100 µL of the enrichments were streaked onto fresh GYPS agar plates and individual colonies growing from these plates after incubation were subcultured onto fresh GYPS agar plates as pure cultures. Visible colonies that appeared floating on top of the liquid media were also subcultured onto fresh GYPS agar plates. These pure cultures were subjected to a molecular identification based on sequencing of the internal transcribed spacer regions of rDNA including the 5.8S (ITS).

## Identification of fungal cultures

Mycelia from 1-week old cultures were scraped from the agar plates and ground into fine powder in liquid nitrogen using a mortar and pestle. Genomic DNA was extracted using the DNeasy Plant DNA Extraction Kit (Qiagen) according to the manufacturer's instructions. The primers ITS4 (`5'-TCCTCCGCTTATTGATATGC-3'`) and ITS5 (`5'-GGAAGTAAAAGTCG TAACAAGG-3'`), amplifying a region from 18S to 28S rDNA covering ITS1, ITS2, and 5.8S regions, were used for PCR [32]. PCR reactions were performed in 25 µL volumes containing *ca*. 20 ng DNA, 0.2 µM of each primer, 12.5 µL Gran Turismo PreMix (Ten Giga BioTech) and 1 µL of the extracted DNA. The amplification cycle consisted of an initial denaturation step of 95˚C for 2 min, followed by 35 cycles of (a) denaturation (95˚C for 30s), (b) annealing (54˚C for 30s) and (c) elongation (72˚C for 30s) and a final 10 min elongation step at 72˚C. The PCR products were analyzed by agarose gel electrophoresis and sent to Genomics BioSci & Tech (New Taipei City, Taiwan) for sequencing using the same PCR primers. The sequences obtained were checked for ambiguity, assembled in MegaX [33] and submitted to the National Center for Biotechnology Information (NCBI) for a nucleotide BLAST search. The ITS sequences of the fungal isolates were deposited in NCBI with the accession numbers given in S1 Table.

## Metabarcoding study

Total DNA was extracted from the freeze-dried sediment samples using Soil DNA Isolation Maxi Kits (Norgen Biotek) according to the manufacturer's instructions. A nested PCR approach was used. The primers NSA3 and NLC2 [34] was used in the first round of PCR. One microliter from the first-round PCR products was used in the second round of PCR. The primers used in the second PCR were ITS1-F_KYO1 [35] and ITS2 [32] with adapter

sequences added on the 5' end of the primers. These primers were found to amplify sequences of the Ascomycota and the Basidiomycota [36].

PCR reactions were performed in 25 μL volumes containing *ca*. 20 ng DNA, 0.2 μM of each primer, 12.5 μL Gran Turismo PreMix (Ten Giga BioTech) and 1 μL first-round PCR product. The amplification cycle consisted of an initial denaturation step of 95˚C for 2 min followed by 35 cycles of (a) denaturation (95˚C for 30s), (b) annealing (54˚C for 30s) and (c) elongation (72˚C for 30s) and a final 10 min elongation step at 72˚C. Five PCR reactions were performed for each sample and pooled. The pooled sample was run on a 1% agarose gel, gel-purified using EasyPure PCR/Gel Extraction Kit (Bioman Scientific Co., Ltd.) and sent to Genomics BioSci & Tech (New Taipei City, Taiwan) for Illumina sequencing (MiSeq Reagent kit v3, 600 cycles). TruSeq DNA Nano (input 120 ng / PCR 5 cycles) was used for library preparation. Negative PCR controls (water controls) were run to detect contamination of the PCR ingredients.

Most of the following bioinformatics processes were run in QIIME 1.9.0 [37]. The raw sequences were filtered with a phred score $\geq$Q29 (a base call accuracy of $\geq$99.87%). The raw reads were merged into single reads and adaptors, primers and barcode sequences were removed using QIIME with the script split_library.py [37]. Clustering was performed using uclust v1.2.22q [38] in QIIME [37]. The reads were processed with UCHIME [39] to remove chimeric sequences. Assigning Operation Taxonomic Units (OTUs) and taxonomic assignments were performed with an open-reference OTU picking approach against the UNITE database in QIIME [37]. A similarity threshold of 97% was adopted. Taxonomic assignment of representative OTUs was run at a 0.97 confidence threshold against the UNITE ITS1 database with UNITE 7.2 reference OTU database ("UNITE+INSD" dataset) using the assignTaxonomy method [40]. The raw reads were deposited in the SRA database with the accession number PRJNA574255.

## Statistical analysis

Rarefaction and extrapolation sampling curves were computed and plotted to estimate sample completeness (sample coverage) in R package iNEXT (iNterpolation/EXTrapolation) with the 95% lower and upper confidence limits for the isolation and metabarcoding data [41]. Principle component analysis (PCA) was used to analyze trends in fungal species composition (based on OTU and percentage normalization of reads) of the sediments (black or yellow) collected from the three collection spots (vent region, east of Turtle Island, west of vent area, Fig 1) between 2015 and 2017 and calculated by R (version 3.6.1) using R studio [42] using package factoextra [43].

Alpha (abundance) and beta (Bray-Curtis similarity) diversities were calculated in R package [42]. To investigate the relationship between species composition and sampling sites, i.e. differences in species composition are smaller for sites that are closer together than for sites that are further apart, a Mantel test (based on Pearson's product-moment correlation coefficient) was conducted to test the null hypothesis that two matrices, spatial distance and ecological distance, are unrelated with alpha = 0.05 in R package [42].

## Results

### Culturable diversity of fungi

A total of 94 isolates was cultured from the crab *Xenograpsus testudinatus*, sediment, seawater and animal egg samples (S1 Table). These fungi were identified based on comparisons of their ITS sequences with those in the GenBank database. The identity of the fungi was referred in most cases to a species name and in some cases to a genus, based on the top BLAST results

with the highest score, in terms of high query coverage (%) and identity (%). When the top BLAST results were unidentified/uncultured fungi, the next result with a name was used if the % query coverage and % identity were high (≥95%); if these figures were low, the identity was only referred to the family, order, class or phylum.

Table 2 and Fig 2 summarize the fungal species richness obtained from the different samples. Sediment samples had the highest species richness: 25 species from the yellow sediment

**Table 2. List of fungal species from the culture-based study.** Fungal species isolated from crab, black sediment, yellow sediment, seawater and animal egg samples collected near/at the hydrothermal vents at Kueishan Island, Taiwan. Fungi isolated from more than one sample type are marked with an asterisk.

| Crab | Black sediment | Yellow sediment | Seawater | Animal eggs |
|------|----------------|-----------------|----------|-------------|
| **Ascomycota** | | | | |
| Capnodiales | Capnodiales | Capnodiales | Capnodiales | Hypocreales |
| *Hortaea werneckii** | *Hortaea werneckii** | *Fodinomyces uranophilus* | *Hortaea werneckii** | *Cordyceps takaomontana* |
| Eurotiales | Eurotiales | *Hortaea werneckii** | Hypocreales | *Fusarium* sp. |
| *Aspergillus sydowii* | *Aspergillus* sp. 2 | Eurotiales | *Parengyodontium album** | |
| *Aspergillus terreus** | *Aspergillus taichungensis* | *Aspergillus aculeatus* | | |
| *Aspergillus unguis* | *Aspergillus terreus** | *Aspergillus terreus** | | |
| *Penicillium citreosulfuratum* | *Penicillium citrinum* | *Penicillium oxalicum* | | |
| Hypocreales | Glomerellales | *Penicillium* sp. | | |
| Hypocreales sp. | *Gibellulopsis nigrescens* | *Penicillium sumatrense* | | |
| *Parengyodontium album** | Hypocreales | Hypocreales | | |
| Microascales | *Parengyodontium album** | *Acremonium brunnescens* | | |
| *Microascus brevicaulis* | *Trichoderma harzianum** | *Acremonium citrinum* | | |
| Saccharomycetales | Microascales | *Acremonium felinum* | | |
| *Candida oceani* | Microascales sp. | *Trichoderma harzianum** | | |
| Xylariales | Pleosporales | Ophiostomatales | | |
| *Hypoxylon monticulosum* | *Allophoma tropica* | *Sporothrix* sp. | | |
| *Peroneutypa scoparia** | *Didymella* sp./*Phoma* sp. | Pleosporales | | |
| *Xylaria* sp. | *Leptosphaeria* sp. | *Curvularia clavata* | | |
| | *Microsphaeropsis arundinis* | *Westerdykella dispersa** | | |
| | Pleosporales sp. | Saccharomycetales | | |
| | *Westerdykella dispersa** | *Meyerozyma guilliermondii* | | |
| | Sordariales | Xylariales | | |
| | *Chaetomium globosum* | *Peroneutypa scoparia** | | |
| | Xylariales | *Xylaria apiculata* | | |
| | *Arthrinium arundinis* | *Xylaria curta* | | |
| | *Arthrinium hydei* | *Incertae sedis* | | |
| | *Incertae sedis* | Ascomycetes sp. | | |
| | Pezizomycotina sp. | | | |
| **Basidiomycota** | | | | |
| Agaricales | Agaricales | Agaricales | | |
| *Chondrostereum* sp.* | *Chondrostereum* sp.* | *Chondrostereum* sp.* | | |
| | Polyporales | Cystobasidiales | | |
| | *Bjerkandera adusta* | *Cystobasidium minutum* | | |
| | *Cerrena* sp.* | Hymenochaetales | | |
| | | *Tropicoporus* sp. | | |
| | | Polyporales | | |
| | | *Cerrena* sp.* | | |
| | | *Earliella scabrosa* | | |
| | | *Porostereum* sp. | | |

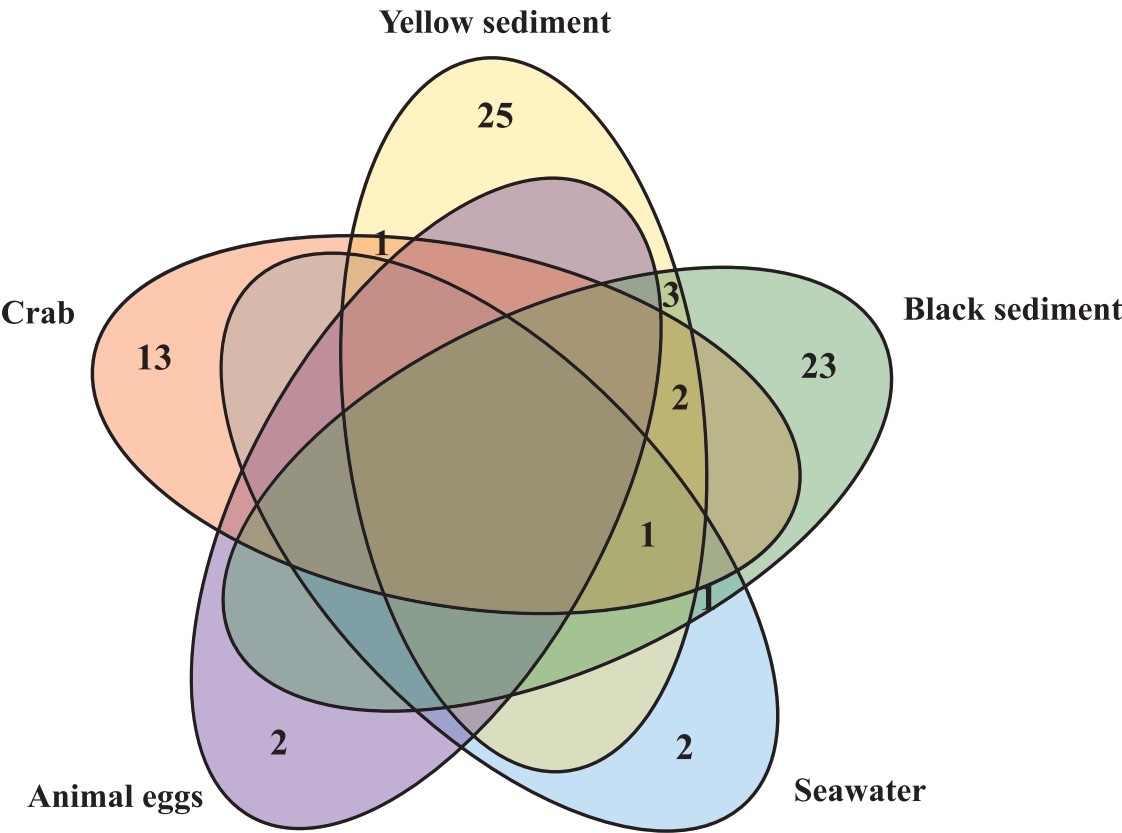

**Fig 2. Venn diagram.** The numbers represent fungal species richness within and between yellow sediment, black sediment, vent crab *Xenograpsus testudinatus*, animal egg and seawater samples. Sediment samples had the highest species richness. *Hortaea werneckii* was the only fungus isolated from four types of samples.

and 23 species from the black sediment. Thirteen species were isolated from the crab samples. Only two species each were isolated from the seawater and the animal egg samples. A higher proportion of the Ascomycota was isolated from the black sediment samples than the yellow sediment samples based on the total species richness (>70%, Fig 3a and 3d). At the class level, 5 different classes of fungi were isolated from the black and 6 from the yellow sediment samples (Fig 3b and 3e). At the order level, the fungi isolated from the yellow sediment could be referred to 11 different fungal orders, followed by black sediment (10), crabs (7), seawater (2) and animal egg (1) samples (Table 2, Fig 3c and 3f). Dominant orders based on species richness were Agaricales, Capnodiales, Eurotiales, Hypocreales, Pleosporales, Polyporales and Xylariales. Species of the Hypocreales were isolated from all types of samples (Table 2).

Concerning the common fungal species between the different sample types, *Hortaea werneckii* was the only fungus isolated from four types of samples, i.e. crab, seawater, yellow and black sediment samples (Table 2). *Parengyodontium album* was isolated from the crab, black sediment and seawater samples. Two species were common between the crab, yellow and black sediment samples (*Aspergillus terreus*, *Chondrostereum* sp.). One common species was isolated from the crabs and the yellow sediment (*Peroneutypa scoparia*). Three other species were isolated from both the black and yellow sediments (*Cerrena* sp., *Trichoderma harzianum*, *Westerdykella dispersa*).

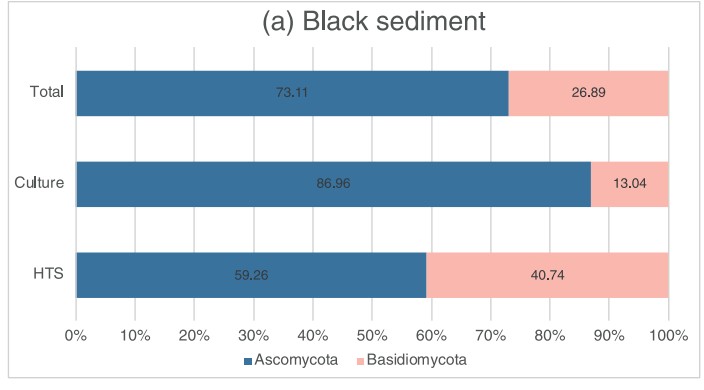

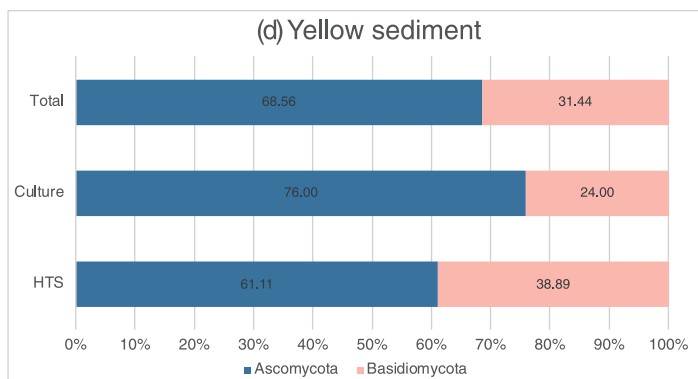

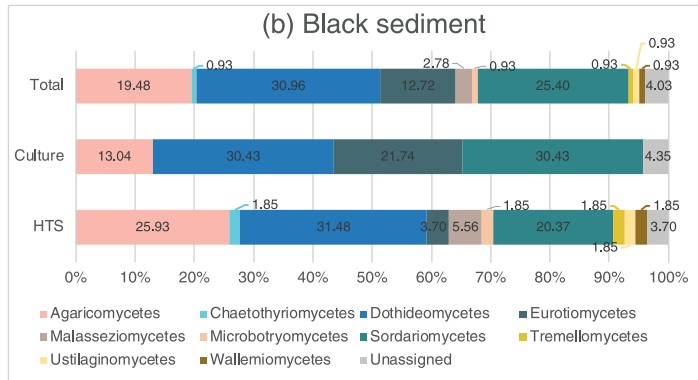

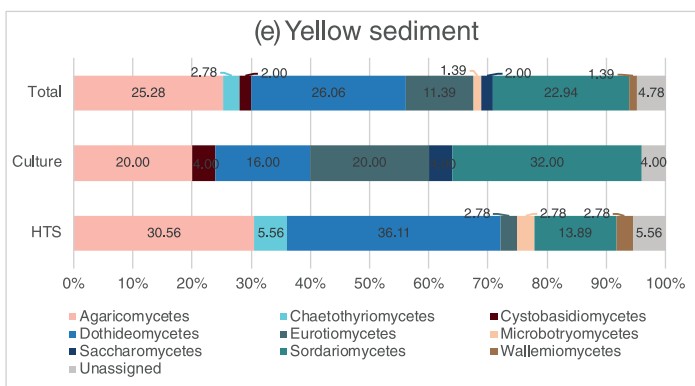

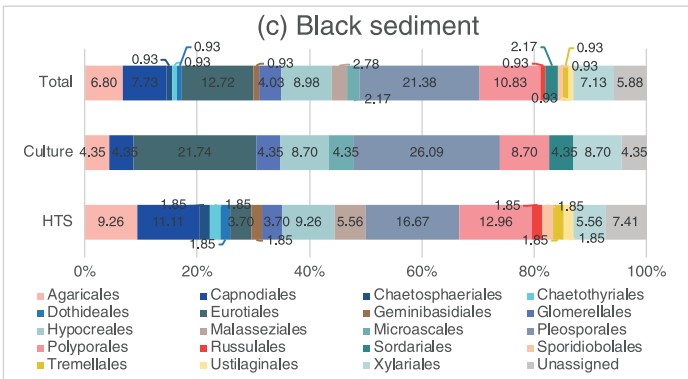

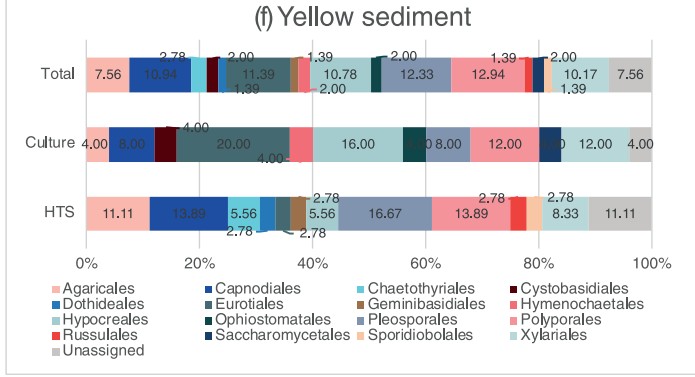

**Fig 3. Diversity of fungi (based on percentage of species richness) in the black and yellow sediment samples.** Classification at phylum, class and order levels using culture-based (Culture) and metabarcoding (HTS) approaches. Total combines the results from both methods. Diversity of fungi at the phylum, class and order levels was different depending on the methods (Culture/HTS) and sample types (yellow/black sediment).

## Metabarcoding analysis of sediments

No PCR products were obtained from the negative PCR controls (water controls). Twenty-two sediment subsamples from the five collection trips were subjected to DNA extraction and the PCR. Positive PCR products were obtained from only 13 sediment subsamples (4 yellow and 9 black sediment subsamples) collected on 23/06/2015, 22-23/08/2016, 29/03/2017 and 28/06/2017 and subsequently sent for the high throughput sequencing. The average read lengths before quality trimming were 1 to 273 base pair (bp) and reads of 271–273 bp were used for the analyses. In S2 Table, the number of raw reads in the 13 subsamples ranged from 56,965 to

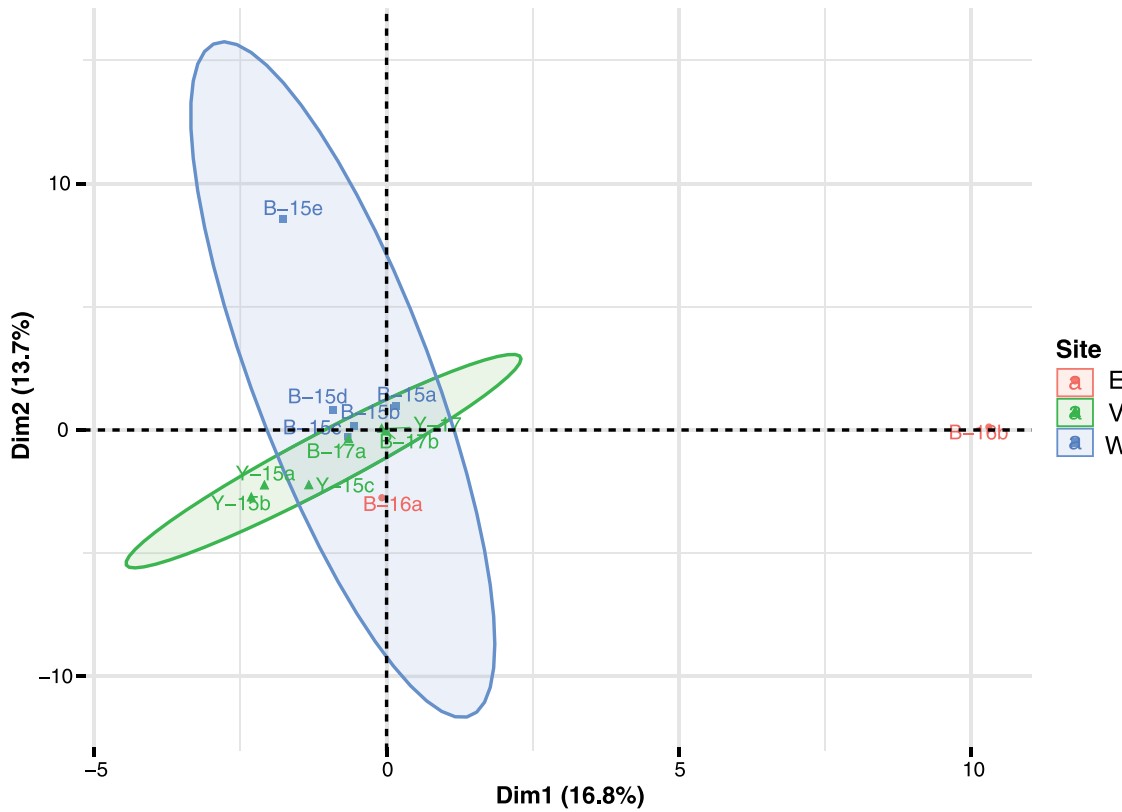

**Fig 4. Principle component analysis (PCA) of the 13 sediment subsamples analyzed with metabarcoding data.** Sample code: V = vent area, E = further away from vent, W = west of vent area; B = black sediment sample, Y = yellow sediment sample; 15 = collected in 2015, 16 = collected in 2016, 17 = collected in 2017; a-e = subsample number. The first axis (PC1) described 16.8% of the variation in the species composition between sediment samples while the second axis (PC2) described 13.7% of the variation, giving a total of 30.5% of the variation in the dataset.

325,131. After removal of chimeric sequences and low quality reads, the number of clean reads in the 13 subsamples ranged from 15,211 to 113,956, with a total of 773,112 reads and an average of 59,470 reads per sample. Majority of the sequences from these samples was of a fungal origin; a low number of sequences (ranged from 41 to 294 reads) was classified as unassigned in 4 out of the 13 subsamples. The PCA analysis based on operational taxonomic units (OTU) and percentage normalization of reads shows that the first axis (PC1) described 16.8% of the variation in the species composition between sediment samples while the second axis (PC2) described 13.7% of the variation, giving a total of 30.5% of the variation in the dataset (Fig 4). From the score plot, sediment samples with a similar species composition clustered together. Generally, there was a gradient of species composition along PC2 axis, decreased from the black sediment samples collected at site W (west of vent area) to site V (vent area), and to site E (further away from vent). PC1 axis contributed significantly to one of the two black sediment samples at site E (B16-b). If the outliner B16-b was removed from the analysis, both PC1 and PC2 contributed significantly to the other black sediment sample at site E (B16-a) (results not shown). Overall, yellow sediment samples of the vent area had the lowest species composition in both PCA1 and PCA2 when compared with other sediment samples (black sediments, regardless of sites).

Reads from the 13 subsamples were combined and referred to 5 samples based on the date of collection and sample type. The rarefaction analysis shown in Fig 5 reveals that species

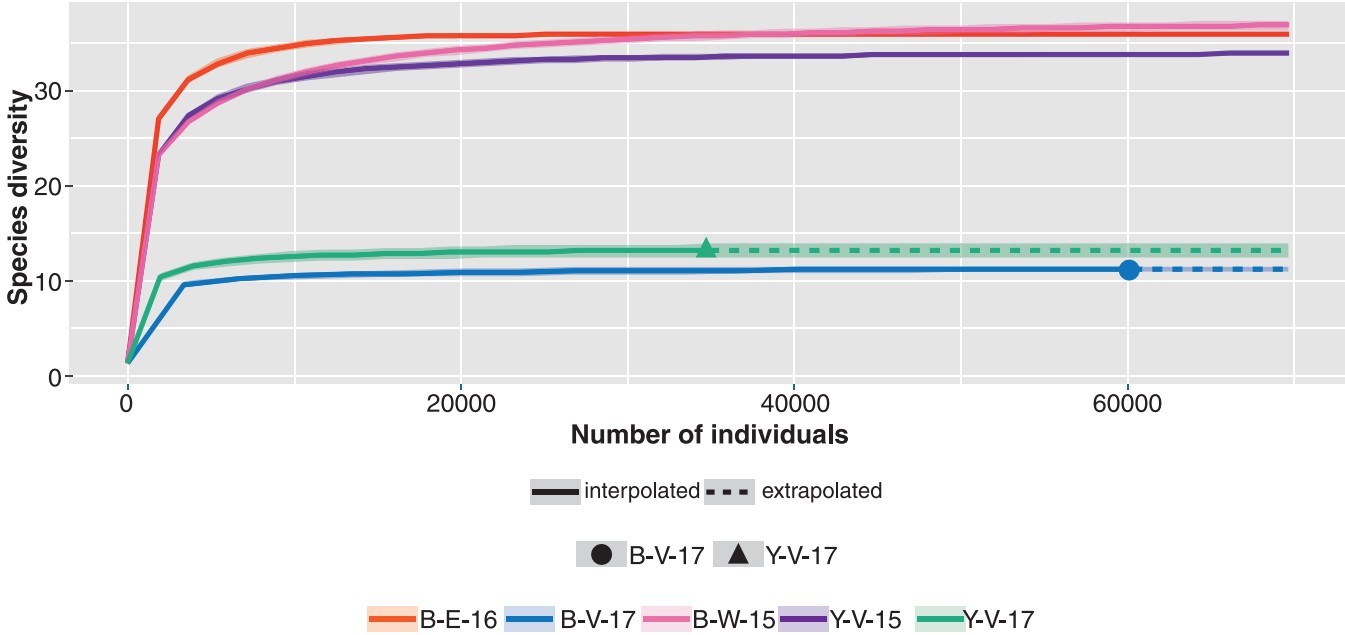

**Fig 5. Rarefaction analysis of the five sediment samples based on sequence reads.** Sample code: B = black sediment sample, Y = yellow sediment sample; V = vent area, E = further away from vent, W = west of vent area; 15 = collected in 2015, 16 = collected in 2016, 17 = collected in 2017. Species diversity of all samples had reached saturation.

diversity had reached saturation for all 5 samples. Fig 6 is a heatmap showing the Bray-Curtis distances between the five sediment samples (beta diversity). The samples of the same type (black, yellow sediment) and collected on the same dates did not group together.

The Mantel test revealed that there was a low correlation among the spatial distance and the ecological distance with negatively association (r = -0.1098) (results not shown). Species composition and sampling sites were concluded to be unrelated ($p = 0.55$).

Excluding the sequences belonging to the Fungi/Unassigned category, species richness of fungi in the black sediment samples was in the range of 7–27 OTUs, in the yellow sediment samples in the range of 12–27 OTUs (species/taxa hereafter) (results not shown). In the black sediment samples, the Ascomycota and the Basidiomycota were equally abundant but the Ascomycota was dominant over the Basidiomycota in the yellow sediment samples (Fig 7a and 7d). At the class and order levels, the fungi were more diverse in the black sediments than the yellow sediments; 10 classes and 17 orders in the black sediments (Fig 7b and 7c) while only 7 classes and 12 orders in the yellow sediments (Fig 7e and 7f). In the yellow sediments, the dominant classes were the Dothideomycetes (56.18% in abundance) and Agaricomycetes (9.49%) while the other classes constituted only small fractions of the sequences (~1%) (Fig 7e). The dominant orders in the yellow sediments were Pleosporales (55.62%) and Polyporales (9.19%) and the rest took up only small percentages (Fig 7f). In the black sediments, the Dothideomycetes (13.09%), the Microbotryomycetes (10.90%) and the Agaricomycetes (7.19%) were the dominant classes (Fig 7b) and the Sporidiobolales (10.90%), the Pleosporales (10.31%) and the Chaetothyriales (4.11%) were the dominant orders (Fig 7c).

## Total diversity of fungi in sediments

Fig 3 shows the diversity of fungi isolated from the black and yellow sediment samples at the phylum, class and order levels from the isolation and metabarcoding methods (excluding

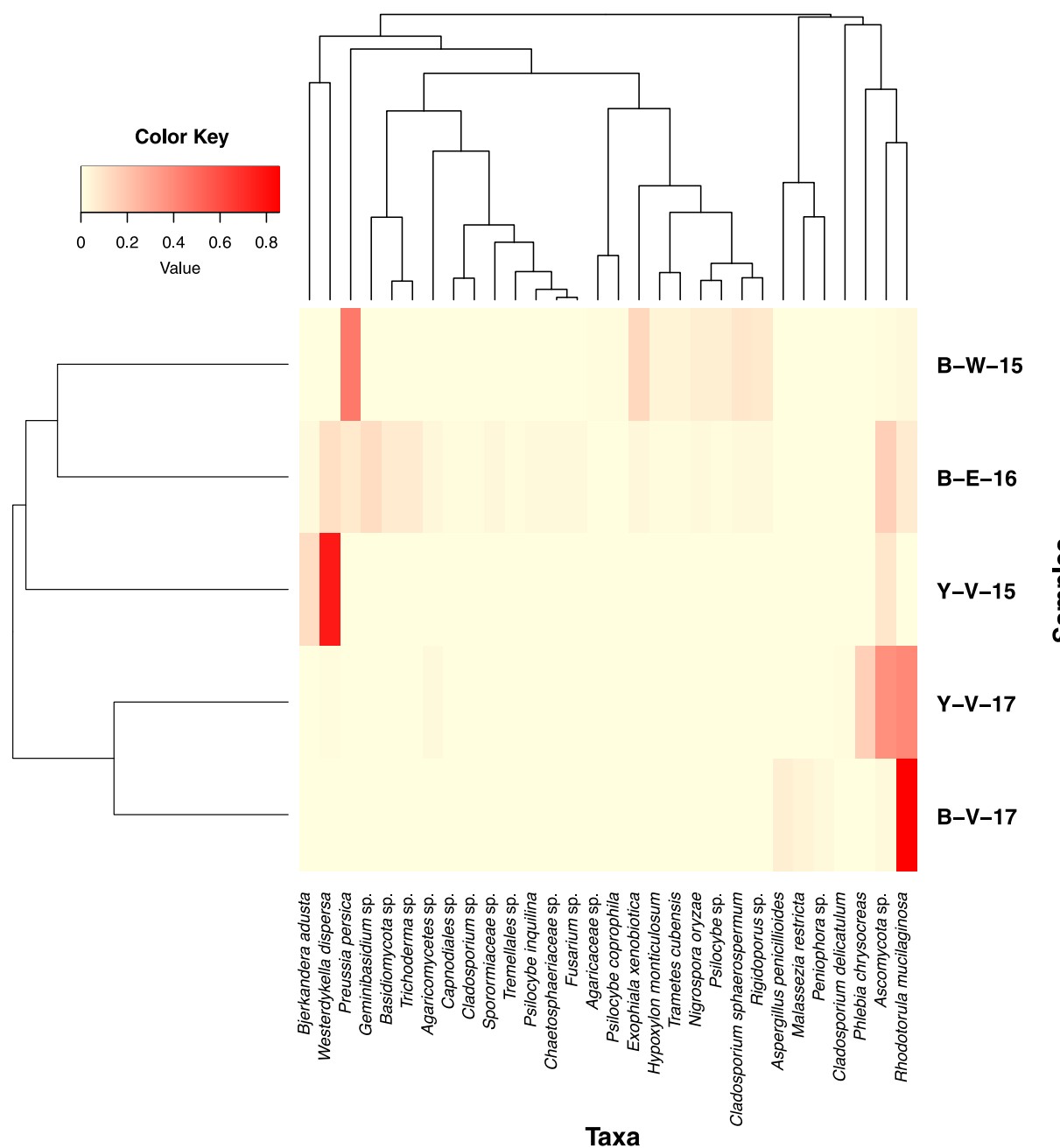

**Fig 6. Heatmap of the Bray-Curtis distances between the five sediment samples (beta diversity).** Heatmap depicting the relative abundance of the 30 most abundant taxa of each samples using Bray-Curtis dissimilarity. Each row in the heatmap represents a specific sample and each column represents a taxon. Colors represent the scaled relative abundance of taxa with light yellow indicating low abundance of that taxon and red as the most abundant. The degree of similarity of mycobiota is represented by the dendrogram on the x-axis and the degree of similarity of samples is shown by the dendrogram on the y-axis. The samples of the same type (black, yellow sediment) and collected on the same dates did not group together. Sample code: B = black sediment sample, Y = yellow sediment sample; V = vent area, E = further away from vent, W = west of vent area; 15 = collected in 2015, 16 = collected in 2016, 17 = collected in 2017.

unidentified/unknown fungi) based on species richness. Generally, the Ascomycota was dominant over the Basidiomycota in the isolation approach (Fig 3a and 3d). More diverse classes and orders, on the other hand, were obtained from the metabarcoding method (Fig 3b, 3c, 3e and 3f). A comparison of the common species in the black and yellow sediments at the class,

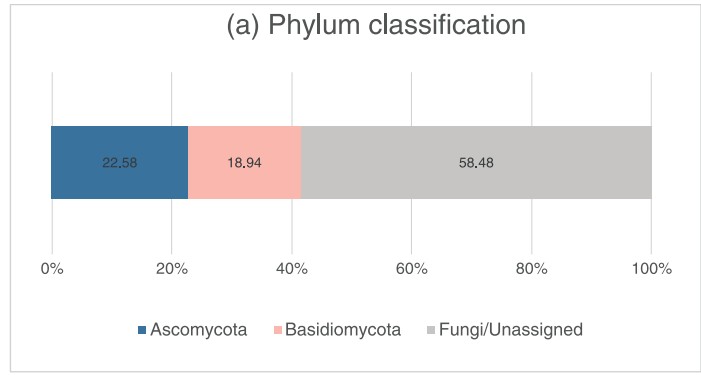
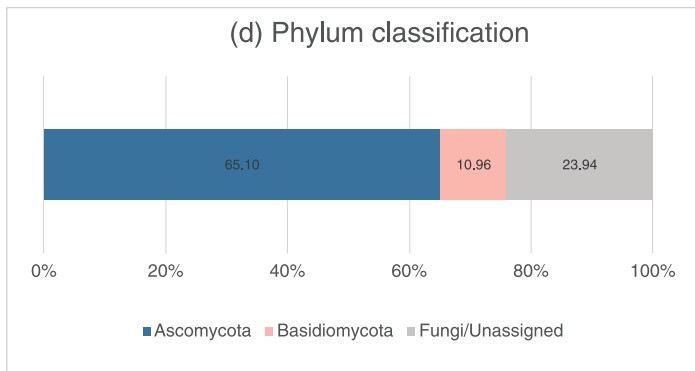
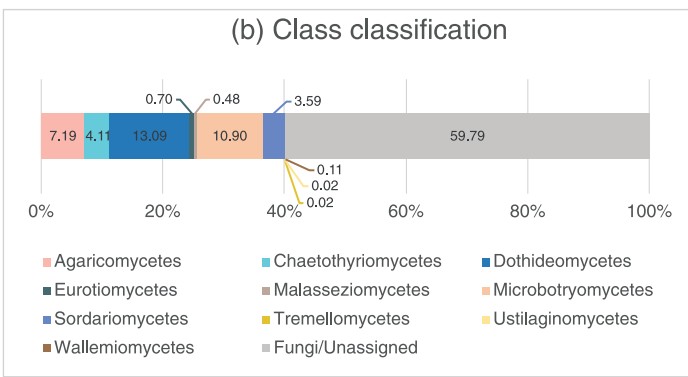
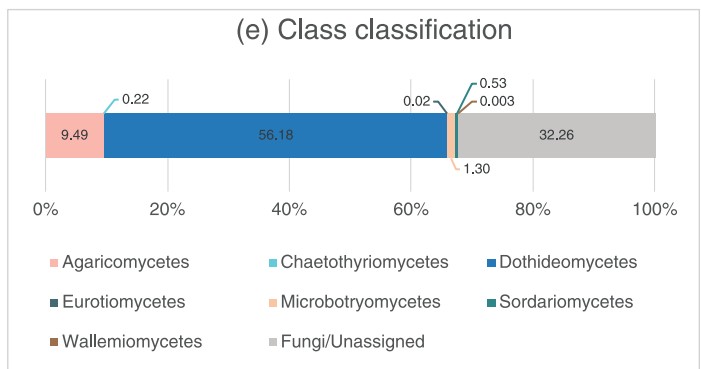
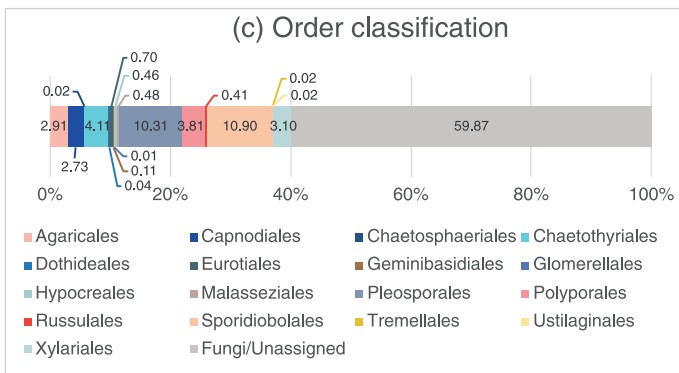
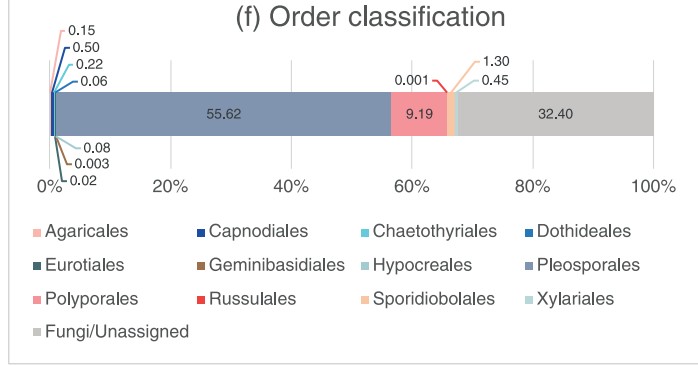

**Fig 7. Diversity of fungi (percentage of reads) in the black (a-c) and yellow (d-f) sediment samples based on the metabarcoding analysis.** Diversity of fungi at phylum, class and order levels was different between the black and yellow sediment.

order and genus levels between the two methods is shown in Fig 8. A higher richness was found in the black sediment samples and in the metabarcoding analysis at the class, order and genus levels.

Overall, the proportion of Ascomycota (~70%) to Basidiomycota in the black and yellow sediment samples was similar (Fig 3a and 3d). At the class level, the fungi in the black sediment samples could be referred to 10 different fungal classes and the dominant classes were Agaricomycetes (Basidiomycota), Dothideomycetes and Sordariomycetes (Ascomycota) (Fig 3b). These three classes among six other classes were also dominant in the yellow sediment samples (Fig 3e). Nineteen different fungal orders were discovered from the black sediment samples (Fig 3c) while sixteen from the yellow sediment samples (Fig 3f). The top five

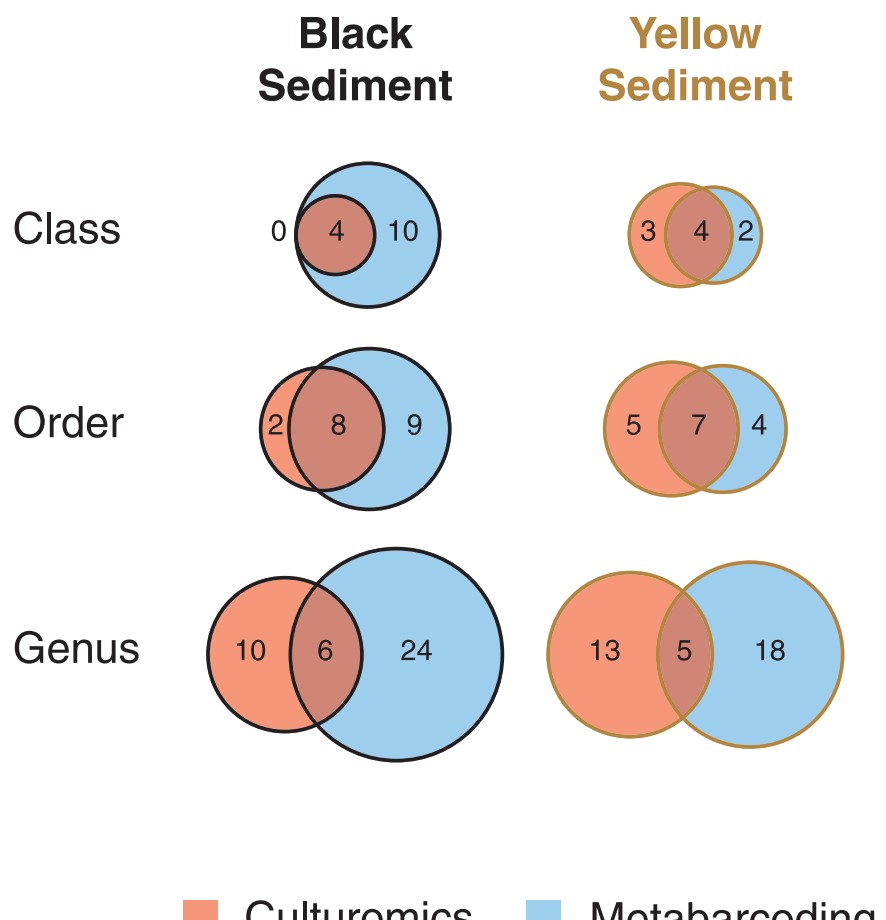

**Fig 8. Fungal richness.** Comparison of fungal richness at class, order and genus levels in black and yellow sediment samples using culture-based method and metabarcoding analysis. A higher richness was found in the black sediment samples and in the metabarcoding analysis at the class, order and genus levels.

dominant fungal orders in descending order based on species richness were Pleosporales>Eurotiales>Polyporales>Hypocreales>Capnodiales in the black sediment samples, and Polyporales>Pleosporales>Eurotiales>Capnodiales>Hypocreales in the yellow sediment samples.

Table 3 lists the species of fungi based on the results of the culture-based and the metabarcoding analyses of the black and yellow sediment samples, excluding those species only classified above the genus level. A total of 54 and 49 species were found in the black and yellow sediments, respectively, excluding the various unknown *Aspergillus* spp. in the black sediment. Twenty-eight species were common between the two sediment types. Species of *Colletotrichum*, *Fusarium* and *Malassezia* were only identified in the black sediment samples, while species of *Acremonium* and *Xylaria* only in the yellow sediment samples.

## Discussion

### Environmental relevance of fungal isolates

The vent crab *Xenograpsus testudinatus*, sediment (yellow and black), seawater and animal eggs were collected at/near the hydrothermal vent system of the Kueishan Island, Taiwan for

**Table 3. List of fungal species in the sediment samples.** The list was constructed based on species from both culture-based and metabarcoding methods. Common species betwen both sediment types are marked with an asterisk.

| Black sediment | Yellow sediment |
|---|---|
| **Ascomycota** | |
| Capnodiales | Capnodiales |
| *Cladosporium delicatulum** | *Cladosporium delicatulum** |
| *Cladosporium sphaerospermum** | *Cladosporium sphaerospermum** |
| *Cladosporium* sp.* | *Cladosporium* sp.* |
| *Hortaea werneckii** | *Hortaea werneckii** |
| Chaetothyriales | *Fodinomyces uranophilus* |
| *Exophiala xenobiotica** | Chaetothyriales |
| Dothideales | *Capronia semi-immersa* |
| *Aureobasidium* sp.* | *Exophiala xenobiotica** |
| Eurotiales | Dothideales |
| *Aspergillus penicillioides** | *Aureobasidium* sp.* |
| *Aspergillus taichungensis* | Eurotiales |
| *Aspergillus terreus** | *Aspergillus aculeatus* |
| *Aspergillus* spp. | *Aspergillus penicillioides** |
| *Penicillium citrinum* | *Aspergillus terreus** |
| Glomerellales | *Penicillium oxalicum* |
| *Colletotrichum brasiliense* | *Penicillium sumatrense* |
| *Colletotrichum gloeosporioides* | *Penicillium* sp. |
| *Gibellulopsis nigrescens* | Hypocreales |
| Hypocreales | *Acremonium brunnescens* |
| *Fusarium solani* | *Acremonium citrinum* |
| *Fusarium* sp. | *Acremonium felinum* |
| *Parengyodontium album* | *Acremonium polychromum* |
| *Simplicillium obclavatum* | *Trichoderma harzianum** |
| *Trichoderma harzianum** | *Trichoderma lixii** |
| *Trichoderma lixii** | Ophiostomatales |
| *Trichoderma* sp. | *Sporothrix* sp. |
| Pleosporales | Pleosporales |
| *Allophoma tropica* | *Curvularia clavata* |
| *Alternaria* sp. | *Preussia persica** |
| *Didymella* sp./*Phoma* sp. | *Pyrenochaetopsis leptospora** |
| *Leptosphaeria* sp. | *Pyrenochaetopsis* sp. |
| *Microsphaeropsis arundinis* | *Sclerostagonospora phragmiticola** |
| *Preussia persica** | *Westerdykella dispersa** |
| *Pyrenochaetopsis leptospora** | Saccharomycetales |
| *Roussoella solani* | *Meyerozyma guilliermondii* |
| *Sclerostagonospora ericae* | Xylariales |
| *Sclerostagonospora phragmiticola** | *Arthrinium* sp.* |
| *Stagonospora neglecta* | *Hypoxylon monticulosum** |
| *Westerdykella dispersa** | *Nigrospora oryzae** |
| Sordariales | *Peroneutypa scoparia* |
| *Chaetomium globosum* | *Xylaria apiculata* |
| Xylariales | *Xylaria curta* |
| *Arthrinium arundinis* | |
| *Arthrinium hydei* | |

*(Continued)*

**Table 3.** (Continued)

| Black sediment | Yellow sediment |
|---|---|
| *Arthrinium* sp.* | |
| *Hypoxylon monticulosum*<sup></sup>* | |
| *Nigrospora oryzae*<sup></sup>* | |
| **Basidiomycota** | |
| Agaricales | Agaricales |
| *Chondrostereum* sp.* | *Chondrostereum* sp.* |
| *Coprinopsis urticicola* | *Psilocybe coprophila*<sup></sup>* |
| *Psilocybe coprophila*<sup></sup>* | *Psilocybe* sp.* |
| *Psilocybe inquilina* | Cystobasidiales |
| *Psilocybe* sp.* | *Cystobasidium minutum* |
| Geminibasidiales | Geminibasidiales |
| *Geminibasidium* sp.* | *Geminibasidium* sp.* |
| Malasseziales | Hymenochaetales |
| *Malassezia globosa* | *Tropicoporus* sp. |
| *Malassezia restricta* | Polyporales |
| Polyporales | *Bbjerkandera adusta*<sup></sup>* |
| *Bjerkandera adusta*<sup></sup>* | *Cerrena* sp.* |
| *Cerrena* sp.* | *Earliella scabrosa* |
| *Phanerochaete tuberculata* | *Phlebia chrysocreas*<sup></sup>* |
| *Phlebia chrysocreas*<sup></sup>* | *Porostereum* sp. |
| *Rigidoporus* sp.* | *Rigidoporus* sp.* |
| *Trametes cubensis*<sup></sup>* | *Trametes cubensis*<sup></sup>* |
| Russulales | Russulales |
| *Peniophora* sp.* | *Peniophora* sp.* |
| Sporidiobolales | Sporidiobolales |
| *Rhodotorula mucilaginosa*<sup></sup>* | *Rhodotorula mucilaginosa*<sup></sup>* |

the isolation of fungi. The sediment samples yielded the highest fungal richness, supporting recent studies revealing complex fungal communities associated with different marine sediments, either shallow [44,45] or deep [19,20]. Marine sediments seem to represent a reservoir of fungi, most of them being ubiquitous [46]. These results may also be linked to the fact that more sediment samples were collected than the other samples, i.e. crabs, seawater and animal eggs and consequently, more species were isolated from the sediment samples.

Thirteen species in seven orders of fungi were isolated from the crab *Xenograpsus testudinatus* with *Aspergillus* as the most retrieved genus on the crab. The association between such fungi and the crab is unclear and an in-depth study is required to get insights into real or erratic association, e.g. by characterizing biofilm on carapace of the crab, by examining more crab samples at the same site to see whether fungi can lead to diseases (necrosis for example) or by examining the ability of fungi isolated on dead crabs to synthesize specific hydrolytic enzymes such as chitinases. It is also possible that the cultures isolated from the crabs (and the animal eggs) represented resting spores on the crab surface at the time of collection.

*Hortaea werneckii* is likely to be a common fungus at/near the hydrothermal vent system at Kueishan Island as it was the only fungus cultured from the sediment, seawater and crab samples. *Hortaea werneckii* is possibly a marine fungus, although it was not listed as a marine species [47]. This species is halophilic [48] and has previously been reported from a hydrothermal vent at the Atlantic Ocean [14] and scuba diving equipment [49]. *Hortaea werneckii*

was reported to cause tinea nigra in humans [50] and may occur as an opportunistic pathogenic fungus but whether this fungus can cause diseases of vent animals requires a thorough investigation.

Seven common orders (Agaricales, Capnodiales, Eurotiales, Hypocreales, Pleosporales, Polyporales, and Xylariales) of fungi were isolated from the yellow and black sediments and six species (*Aspergillus terreus*, *Cerrena* sp., *Chondrostereum* sp., *Hortaea werneckii*, *Trichoderma harzianum*, *Westerdykella dispersa*) were cultured from both sample types. Fungi of the Agaricales, Capnodiales, Eurotiales, Hypocreales, and Polyporales are common in the marine environment, especially seawater and sediment [47]. The Xylariales has been isolated as endophytes of marine-adapted mangrove plants, seagrasses and macroalgae [51]. However, *Peroneutypa scoparia* of the Xylariales was reported to be an endophyte of plants and also a wood inhabitant [52] but it is unknown in the marine environment. No yeasts were isolated from the black sediment and only two marine yeast species, *Cystobasidium minutum* and *Meyerozyma guilliermondii*, were isolated from the yellow sediment. Yeasts are common in the marine environment, especially basidiomycetous yeasts [53] and the low number of yeasts isolated may be related to the isolation methods. Yeasts were covered by dense mycelia of filamentous fungi on the isolation plates which impeded their isolation. Sequences of the yeasts *Geminibasidium* sp. and *Rhodotorula mucilaginosa* recovered through the metabarcoding analysis confirm the presence of other yeast species in both sediment types.

## Molecular diversity of fungi

Ascomycota and Basidiomycota were the only phyla identified from sequences of the metabarcoding analysis. No basal fungal lineages, such as the Chytridiomycota and the Cryptomycota, were obtained, although such lineages are common in the marine environment [54]. This might be due to the primer bias towards the amplification of the Ascomycota and the Basidiomycota in the PCR reactions.

The fungal communities of the five sediment samples analyzed through the metabarcoding analysis did not form separate clusters based on the date of collection and sample types, suggesting that no temporal variation or substrate-specificity seemed to occur. However, the whole picture (Fig 3) strongly highlights that each type of sediment harbors specific fungal communities.

Over half of the total reads in the yellow sediment belonged to the Pleosporales (55.62%) of the Dothideomycetes (56.18%) due to the high number of reads belonging to *Westerdykella dispersa* (55.11%). The role of *W. dispersa* near the hydrothermal vent is unknown but this species was previously reported from the marine sediment of the South China Sea [55]. The proportion of the different dominant classes (Agaricomycetes, Microbotryomycetes, Dothideomycetes, ~10%) and orders (Sporidiobolales, Pleosporales, ~10%) in the black sediment was comparable.

On the sediment samples in which both the isolation and metabarcoding methods were applied, no common species of fungi were recovered from both methods from the black sediment samples collected on 22/08/2016 and 29/03/2017 and the yellow sediment samples collected on 28/06/2017. Culture-dependent methods favor fast-growing fungi such as *Asperigllus* spp. and *Penicillium* spp. (Table 2), species that might not be dominant in the samples. The diverse classes and orders of fungi recovered from the metabarcoding method confirm the pitfalls of using isolation as the sole method to studying the diversity of fungi.

## Ecology of fungi

Based on the results from the metabarcoding analysis and the culture-based approach, the number of fungal species from the yellow (49) and black (54) sediments was comparable.

Among these fungi, the majority have been classified as terrestrial taxa while a few can be recognized as marine [47] (Table 3). Several species found in this study are common species of the deep-sea, including *Penicillium citrinum*, *Meyerozyma guilliermondii* and *Rhodotorula mucilaginosa* [21]. Species richness of the Ascomycota was higher than the Basidiomycota, supporting the results of other studies highlighting the dominance of ascomycetes in the marine environment. Majority of the fungi isolated from animal, sediment, rock and seawater samples collected at various deep sea hydrothermal vent sites in the Pacific and Atlantic Oceans belonged to Coniochaetales, Hypocreales, Helotiales, Chaetothyriales and Eurotiales of the Ascomycota while only one basidiomycete was obtained [13]. On sulfide and black smoker samples collected at a deep-sea hydrothermal vent site located near the Mid-Atlantic Ridge of the South Atlantic Ocean, 129 isolates belonging to the Ascomycota (*Cladosporium*, *Phoma*, *Phialemonium*, *Stachybotrys*, *Penicillium*, *Aspergillus*, *Phialophora*, *Botryotinia*, *Meyerozyma*, unclassified Hypocreales and unclassified Xylariaceae) and 32 isolates to the Basidiomycota (*Rhodotorula*, *Tilletiopsis* and *Sporobolomyces*) were isolated [55]. However, basidiomycetous yeasts (e.g. *Cryptococcus*, *Rhodosporidium*, *Rhodotorula*) were found predominantly on Fe-oxide mats and basalt rock surfaces at the active volcano, Vailulu'u seamount, Samoa [56]. The Chytridiomycota was not recovered in this study by both culture and metabarcoding methods but sequences of this phylum were obtained from animal and rock samples of deep-sea hydrothermal vents at the East Pacific Rise and the Mid-Atlantic Ridge using PCR-cloning-sequencing analysis [22].

The dominant classes (i.e. Agaricomycetes, Dothideomycetes and Sordariomycetes) and orders (i.e. Capnodiales, Eurotiales, Hypocreales, Pleosporales, Polyporales, Xylariales) in both the yellow and black sediment types were similar. Most marine Pleosporales are intertidal mangrove species [47, 57,58]. As discussed above, basidiomycetous yeasts are common in the marine environment but not many filamentous basidiomycetes are known in the marine environment [47] and it may be related to salinity sensitivity [59]. Occurrence of filamentous basidiomycetes in this report and in other studies may suggest that spores of the terrestrial Polyporales (e.g. *Bjerkandera adusta*, *Cerrena* sp., *Phanerochaete tuberculate*, *Phlebia chrysocreas*, *Rigidoporus* sp., *Trametes cubensis* and *T. versicolor*) and Agaricales (e.g. *Chondrostereum* sp. and *Schizophyllum commune*) deposit in the marine sediment [19,20]. Kueishan Island is only ~10 km away from the main Taiwan Island and it is likely that spores of the filamentous basidiomycetes were originated from the terrestrial environment through freshwater runoff or air deposition [60] and deposited into the sediment. Four species are described in *Chondrostereum* and *C. purpureum* can cause the silverleaf disease, which causes the death of plants and its spores are dispersed by air [61]. Fröhlich-Nowoisky et al. [60] examined the diversity of fungi in continental, coastal and marine air; the Basidiomycota constituted 41% and 28% of the total fungal propagules in coastal and marine air, respectively and the dominant class was Agaricomycetes in both air types. It is likely that spores of the Agaricomycetes settle into the ocean and eventually sink to the seabed but they play no role in the sediment. The shallow nature of the Kueishan Island vent field and its close proximity to land strongly support the idea of terrestrial spore/hyphae dissemination from land to the marine environment. Although speculative, such preliminary results pave the road for a complementary fungal-focused metatranscriptomic approach to delve deeper into marine fungal functions at the mRNA expression level. In the same study [60], the Ascomycota belonging to the Dothideomycetes, Eurotiomycetes, Leotiomycetes and Sordariomycetes was dominant in the coastal and marine air and it may also explain the dominance of the Dothideomycetes, Eurotiomycetes, and Sordariomycetes in the sediment.

Species of *Colletotrichum*, *Fusarium* and *Malassezia* were only identified in the black sediment samples, the same is true for species of *Acremonium* and *Xylaria* in the yellow sediment

samples. *Colletotrichum* is a genus of common plant disease fungi including a range of agricultural plants and of endophytes of terrestrial plants [62]. Currently available evidence does not suggest *Colletotrichum* play a role in the marine sediment, same for *Xylaria* spp. Members of the genus *Fusarium* are also well known in terrestrial habitats as common soil fungi or plant pathogens. However, some *Fusarium* species have true roles in the marine environment; for example *F. oxysporum* has been reported to associate to marine mammals like dolphins and whales [63], causing dermatitis and systemic lesions sometimes leading to mortalities, and also known as a denitrifying species in oxygen-depleted regions in the marine environment [64]. Species of the genus *Malassezia* can cause skin diseases of human [65] and are among the most widespread fungi in the marine environment. DNA signatures of *Malassezia* and *Malassezia*-like sequences have been retrieved from numerous contrasted marine habitats, including sponges and deep-sea sediments [10]. The detection of DNA signatures of *Malassezia* in the black sediment at the hydrothermal vents of Kueishan Island is another proof of *Malassezia* occurrence in the marine environment. Complementary rRNA biosignatures have revealed metabolic activities of this genus in deep-sea sediments [15, 18]. *Malassezia* members may cause diseases of marine animals and represent a relevant target taxon to isolate into culture in the coming years to better assess their ecophysiological features.

This study, as in a number of other studies on fungal diversity of marine sediment, recovered a number of fungal species with potential animal pathogenicity, including *Acremonium* spp., *Aspergillus* spp., *Fusarium* spp., *Malassezia* spp., *Hortaea werneckii*, *Parengyodontium album*, and *Westerdykella dispersa*. Whether *Hortaea werneckii*, *P. album* and *W. dispersa* [66–68] can cause diseases of the vent animals (i.e. the vent crab *Xenograpsus testudinatus*) is unknown at Kueishan Island. More research is required to determine whether any of the fungi recovered from the marine sediment can cause diseases of bottom-dwelling animals and impact this specific food web.

## Conclusions

Fungal diversity in shallow hydrothermal vent system is a gap of our knowledge of the marine mycota. This is the first study to report a high diversity of fungi (54 and 49 species of the Ascomycota and the Basidiomycota) in the black and yellow sediments collected at/near the shallow hydrothermal vent area of Kueishan Island by culture-based and metabarcoding methods. Some of the recovered fungi might be of a terrestrial origin while some may cause diseases of marine animals.

## Supporting information

**S1 Table. BLAST search results of the fungi isolated from sediment, seawater, the vent crab *Xenograpsus testudinatus* and animal egg samples collected at the shallow hydrothermal vent field, Kueishan Island, Taiwan.**
(XLSX)

**S2 Table. Identification of reads/sequences from results of the metabarcoding analysis.**
(XLSX)

## Author Contributions

**Conceptualization:** Ka-Lai Pang.

**Data curation:** Sheng-Yu Guo, I-An Chen, Tsz-Wai Ho, Ling-Ming Tsang, Michael Wai-Lun Chiang.

**Formal analysis:** Ka-Lai Pang, Sheng-Yu Guo, I-An Chen, Gäetan Burgaud, Zhu-Hua Luo, Tsz-Wai Ho, Ling-Ming Tsang, Michael Wai-Lun Chiang, Hyo-Jung Cha.

**Funding acquisition:** Ka-Lai Pang.

**Investigation:** Ka-Lai Pang, Sheng-Yu Guo, I-An Chen, Yi-Li Lin, Jian-Shun Huang, Michael Wai-Lun Chiang, Hyo-Jung Cha.

**Methodology:** Ka-Lai Pang, Sheng-Yu Guo, I-An Chen, Gäetan Burgaud, Hans U. Dahms, Jiang-Shiou Hwang, Yi-Li Lin, Jian-Shun Huang, Tsz-Wai Ho, Ling-Ming Tsang, Hyo-Jung Cha.

**Project administration:** Ka-Lai Pang, Hans U. Dahms, Jiang-Shiou Hwang.

**Software:** Tsz-Wai Ho.

**Supervision:** Ka-Lai Pang, Hans U. Dahms, Jiang-Shiou Hwang.

**Visualization:** Michael Wai-Lun Chiang.

**Writing – original draft:** Ka-Lai Pang, Gäetan Burgaud, Zhu-Hua Luo.

**Writing – review & editing:** Ka-Lai Pang, Gäetan Burgaud, Zhu-Hua Luo.

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
