## [Decision Letter · Decision Letter 0]

21 Aug 2019

PONE-D-19-20402

Insights into fungal diversity of a shallow-water hydrothermal vent field at Kueishan Island, Taiwan by culturomics and metabarcoding analysis

PLOS ONE

Dear Dr. Pang,

Thank you for submitting your manuscript to PLOS ONE. After careful consideration, we feel that it has merit but does not fully meet PLOS ONE’s publication criteria as it currently stands. Therefore, we invite you to submit a revised version of the manuscript that addresses the points raised during the review process.

I received two reviews from experts in the field, and both reviewers agree that the work has value and will be a nice contribution to the literature. However, both reviewers also noted that the manuscript needs to be improved and elaborated prior to acceptance for publication. I am in agreement with them. As pointed out by reviewer #1, the manuscript would be stronger if there was connection to the environmental data. The amplicon data needs to be presented properly. I also highlighted a few area that requires clarity, discussion and revision (below). Also the statistical analysis, together with the amplicon data should be analyzed sufficiently. It is essential to deposit the raw reads in SRA. The manuscript would be improved if you pay attention to these recommendations. 

We would appreciate receiving your revised manuscript by Oct 05 2019 11:59PM. To enhance the reproducibility of your results, we recommend that if applicable you deposit your laboratory protocols in protocols.io, where a protocol can be assigned its own identifier (DOI) such that it can be cited independently in the future. For instructions see: http://journals.plos.org/plosone/s/submission-guidelines#loc-laboratory-protocols

We look forward to receiving your revised manuscript.

Kind regards,

Kin Ming Tsui

Academic Editor

PLOS ONE

Journal Requirements:

3. We note that you are reporting an analysis of a microarray, next-generation sequencing, or deep sequencing data set. PLOS requires that authors comply with field-specific standards for preparation, recording, and deposition of data in repositories appropriate to their field. Please upload these data to a stable, public repository (such as ArrayExpress, Gene Expression Omnibus (GEO), DNA Data Bank of Japan (DDBJ), NCBI GenBank, NCBI Sequence Read Archive, or EMBL Nucleotide Sequence Database (ENA)). In your revised cover letter, please provide the relevant accession numbers that may be used to access these data. For a full list of recommended repositories, see http://journals.plos.org/plosone/s/data-availability#loc-omics or http://journals.plos.org/plosone/s/data-availability#loc-sequencing.

Additional Editor Comments:

line 215-216 - although the amplicons were sent to a commercial platform, it was essential to describe the library preparation method and the package for sequencing (e.g. Miseq V2 or V3). How many cycles of sequencing reaction?

line 231-242 - the description of statistical analysis was inadequate and incomplete. Use R packages or other statistical software instead of MS Excel; Different analysis packages/tools should be used for culture-based and sequence based data. Have you estimated the alpha, beta diversity (similar to Bray-Curtis dissimilarity) for the  amplicon data?

line 295-296 - report the read and sequence information: read length before and after quality trimming; # of chimera reads; # of reads due to contamination (there was a negative control)

line 296 - there was 10 fold difference in # of reads among samples; have you normalized the # of reads before the data analysis. Did you observe any changes in the communities with or without normalization of reads?

Fig.4, line 440-444 - sample B15-b was an outlier.  What could be the possible ecological factors or technical reasons? Another PCA diagram should be generated without B15-b to show the variation among other samples (The current Fig 4 was not acceptable). Otherwise the variations (temporal or substrate specificity) could be masked. 

Fig.6 - the samples were collected from 3 locations; How much observed variation among the communities was due to geographic factor? Mantel test or a correlation between community variation and geographic distance should be established.

line 128, 438 - if the primers were biased towards the higher fungi and if the study did not aim at lower fungi, this should be mentioned in the introduction/ materials and methods.

line 198-199 - the ITS sequences are deposited; however the raw reads also need to be deposited in the GenBank / SRA.

Reviewers' comments:

Reviewer's Responses to Questions

**Comments to the Author**

1. Is the manuscript technically sound, and do the data support the conclusions?

Reviewer #1: Partly

Reviewer #2: Yes

2. Has the statistical analysis been performed appropriately and rigorously? 

Reviewer #1: No

Reviewer #2: Yes

3. Have the authors made all data underlying the findings in their manuscript fully available?

Reviewer #1: Yes

Reviewer #2: Yes

4. Is the manuscript presented in an intelligible fashion and written in standard English?

Reviewer #1: Yes

Reviewer #2: Yes

5. Review Comments to the Author

Reviewer #1: Pang et al. presented a study on the diversity of marine fungi at a shallow hydrothermal vent system, using both cultivation and metabarcoding methods. Their findings will make an important contribution to our knowledge about the biogeography of marine fungi, and should be of interest to microbial ecologists and marine biologists. However, there are several major issues that need to be addressed before it is suitable for publication. In sum I recommend a major revision.

Major comments

The fungal diversity reported from this shallow hydrothermal vent system is valuable, but the impact of the paper can be significantly elevated by simply incorporating relevant environmental parameters such as temperature, depth, oxygen concentration, sulphide concentration, sediment porosity, water turbidity, primary productivity, etc. Are any of these variables measured at the site at the time of sample collection? If not, is there any historic data reported in the literature that can be used for interpreting the fungal diversity?

One major parameter that was repeatedly referred to in the manuscript is the sediment type (yellow vs. black). What caused the sediment to have different colors? How do the physical and chemical properties of the two types of sediments compare? Do they have the same porosity? What is the nutrient concentration in the sediment porewater? What is the depth of each sediment sampled? Similarly, what are the differences between the three sampling sites? Is the yellow sediment from location V the same as the yellow sediment in location E? The multiple locations of sampling are a strength of this study, and it will be of high interest for microbial ecologists to learn which factor played the most important role in determining the fungal community. Was it the sediment type or the physicalchemical variables associated with each location? Specifically, the authors can perform statistical tests to address this question.

Moreover, the discussion of the manuscript is inadequate. Since other marine systems have been mentioned in the introduction, it is natural and necessary to compare the fungal diversity discovered at the shallow hydrothermal vent to other systems such as deep hydrothermal vents. At which location is fungal diversity higher? How important is depth and organic matter input in determining the diversity of marine fungi?

The introduction in its current form requires major revision. The lengthy second paragraph of the introduction is a brief review of deep-sea fungi including those from deep-sea hydrothermal vents, which has little relevance to the current study. I suggest removing the majority of the contents from this paragraph and condense it to a summary of marine habitats where marine fungal diversity has been surveyed. At the end it should be pointed out that fungal diversity in shallow hydrothermal vent system is a gap in our knowledge.

Additionally, the method and/or the result section needs to include a few key pieces of information. As for the metabarcoding analysis of sediments, how many reads were merged? How many reads can be classified as fungal? How many reads were classified as other known taxa? How many reads cannot be classified as any known taxa? What is the classification threshold? As for PCA, how were data transformed before ordination? What is the definition of species composition?

The discussion section should include comparison with other studies on deep-sea hydrothermal vents as mentioned above, and a discussion on what environmental variables may have played a role in determining the fungal diversity. The current section “Putative ecological roles” is a misnomer because the majority of this section deals with either diversity alone or dispersal mechanisms. The section on Malassezia is really about its biogeography and not about its ecological roles. Therefore, I suggest restructuring the section into biogeography and dispersal mechanisms.

Minor comments

“taxa” (line 33) is a vague term and should be avoided when possible. In the manuscript, taxa refer to species/genus in most occasions, and should be revised accordingly.

Line 64-65: it is misleading to state “especially in the deep sea”. I suggest removing this part.

Line 135: (The section on Collection of Samples). Table 1 should include information about which site and which type of substrate was sampled for cultivation.

Line 220: “paired” should be “merged”.

Line 256: “taxa” should be “species”.

Table 2: what does the asterisk sign mean? It needs to be explained in the caption.

Figure 3: it needs to be specified which sub-figures are for cultivation results and which sub-figures are for metabarcoding results.

Reviewer #2: This is nicely written paper on metabarcoding and culture based studies of shallow water hydrothermal vent field in Taiwan.

1. Please don't use the term culturomics! I would use culture-based through out the manuscript instead.

2. Additional coorections are annotated on the attached manuscript.

3. Please include a summary or conclusion section at the end if possible.

6. PLOS authors have the option to publish the peer review history of their article (what does this mean?). If published, this will include your full peer review and any attached files.

Reviewer #1: No

Reviewer #2: No

---

## [Author Response · Author response to Decision Letter 0]

30 Oct 2019

4 October 2019

Clement Kin-Ming Tsui

Associate Editor

PLoS One

Dear Dr. Tsui,

We are submitting a revised manuscript entitled ‘Insights into fungal diversity of a shallow-water hydrothermal vent field at Kueishan Island, Taiwan by culture-based and metabarcoding analysis’. We have attended to all comments made by you and the reviewer and made corresponding changes to the manuscript. We have also reformat the manuscript based on the journal requirements. Here we list the response to the comments:

Associate Editor

Comment 1: ‘line 215-216 - although the amplicons were sent to a commercial platform, it was essential to describe the library preparation method and the package for sequencing (e.g. Miseq V2 or V3). How many cycles of sequencing reaction?’

Response: TruSeq DNA Nano (input 120 ng / PCR 5 cycles) was used for library preparation. Miseq v3 was used and 600 cycles was used in the sequencing reaction.

Comment 2: ‘line 231-242 - the description of statistical analysis was inadequate and incomplete. Use R packages or other statistical software instead of MS Excel; Different analysis packages/tools should be used for culture-based and sequence based data. Have you estimated the alpha, beta diversity (similar to Bray-Curtis dissimilarity) for the amplicon data?’

Response: Figure 6 has been replaced by a new figure showing a heatmap of the Bray-Curtis distances between the five sediment samples (beta diversity).

Comment 3: ‘line 295-296 - report the read and sequence information: read length before and after quality trimming; # of chimera reads; # of reads due to contamination (there was a negative control)’

Response: The average read lengths before quality trimming were 1 to 273 base pair (bp) and reads of 271-273 bp used for the analyses. Control PCRs (water control) were performed but no PCR products were obtained and therefore, these samples were not sequenced for comparison. These information is added to the new Supplementary Table 2 and the text.

Comment 4: ‘line 296 - there was 10 fold difference in # of reads among samples; have you normalized the # of reads before the data analysis. Did you observe any changes in the communities with or without normalization of reads?’

Response: We agree with the editor that normalization is required for the data due to the fold differences between samples. We have re-run the PCR analysis with percentage normalization and Figure 4 has been replaced.

Comment 5: ‘Fig.4, line 440-444 - sample B15-b was an outlier. What could be the possible ecological factors or technical reasons? Another PCA diagram should be generated without B15-b to show the variation among other samples (The current Fig 4 was not acceptable). Otherwise the variations (temporal or substrate specificity) could be masked.’

Response: The PCA was re-run with normalization of reads including the sample B15-b. B15-b clusters with other samples. The outlier is B16-b but only in PC1 which only constitutes 16.8% of the variation. Figure 4 has been replaced.

Comment 6: ‘Fig.6 - the samples were collected from 3 locations; How much observed variation among the communities was due to geographic factor? Mantel test or a correlation between community variation and geographic distance should be established.’

Response: A Mantel test was run and was found that there was a low correlation among the spatial distance and the ecological distance with negatively association (r = -0.1098). Species composition and sampling sites were concluded to be unrelated (p = 0.55). These information has been added to the text.

Comment 7: ‘line 128, 438 - if the primers were biased towards the higher fungi and if the study did not aim at lower fungi, this should be mentioned in the introduction/ materials and methods.’

Response: This has been added in the Materials and Methods and it reads ‘These primers were found to amplify sequences of the Ascomycota and the Basidiomycota.’

Comment 8: ‘line 198-199 - the ITS sequences are deposited; however the raw reads also need to be deposited in the GenBank / SRA.’

Response: The ITS sequences from the isolation study and the raw reads from the metabarcoding study have been deposited in GenBank/SRA and their accession numbers have been added to the manuscript.

Reviewer 1

Comment 9: ‘The fungal diversity reported from this shallow hydrothermal vent system is valuable, but the impact of the paper can be significantly elevated by simply incorporating relevant environmental parameters such as temperature, depth, oxygen concentration, sulphide concentration, sediment porosity, water turbidity, primary productivity, etc. Are any of these variables measured at the site at the time of sample collection? If not, is there any historic data reported in the literature that can be used for interpreting the fungal diversity? One major parameter that was repeatedly referred to in the manuscript is the sediment type (yellow vs. black). What caused the sediment to have different colors? How do the physical and chemical properties of the two types of sediments compare? Do they have the same porosity? What is the nutrient concentration in the sediment porewater? What is the depth of each sediment sampled? Similarly, what are the differences between the three sampling sites? Is the yellow sediment from location V the same as the yellow sediment in location E? The multiple locations of sampling are a strength of this study, and it will be of high interest for microbial ecologists to learn which factor played the most important role in determining the fungal community. Was it the sediment type or the physicalchemical variables associated with each location? Specifically, the authors can perform statistical tests to address this question.’

Response: The characteristics of the sediment samples have been added in the text and it reads ‘Visible sulfur granules were found in the yellow sediment samples (with a higher SO3 content) but not in the black sediment samples (with a lower SO3 content).’ The depth of the sampling sites was also provided in Table 1. However, the other information regarding environmental variable and sample properties mentioned by the reviewer is not available from us or literature.

Comment 10: ‘Moreover, the discussion of the manuscript is inadequate. Since other marine systems have been mentioned in the introduction, it is natural and necessary to compare the fungal diversity discovered at the shallow hydrothermal vent to other systems such as deep hydrothermal vents. At which location is fungal diversity higher? How important is depth and organic matter input in determining the diversity of marine fungi?’

Response: A discussion between the results of this study and others from the deep-sea has been added. Most studies of fungal diversity of hydrothermal vent ecosystem are from the deep-sea and this, to our knowledge, is the first study of fungal diversity of a shallow hydrothermal vent ecosystem. More information on the latter is needed to make a comparison between the two types of hydrothermal vents. Currently, little information is available to relate depth and organic matter input on the diversity of marine fungi. 

Comment 11: ‘The introduction in its current form requires major revision. The lengthy second paragraph of the introduction is a brief review of deep-sea fungi including those from deep-sea hydrothermal vents, which has little relevance to the current study. I suggest removing the majority of the contents from this paragraph and condense it to a summary of marine habitats where marine fungal diversity has been surveyed. At the end it should be pointed out that fungal diversity in shallow hydrothermal vent system is a gap in our knowledge.’

Response: The introduction has been reorganized as suggested by the reviewer. It now only summarizes brief results of fungal diversity from other deep-sea hydrothermal vent studies. 

Comment 12: ‘Additionally, the method and/or the result section needs to include a few key pieces of information. As for the metabarcoding analysis of sediments, how many reads were merged? How many reads can be classified as fungal? How many reads were classified as other known taxa? How many reads cannot be classified as any known taxa? What is the classification threshold? As for PCA, how were data transformed before ordination? What is the definition of species composition?’

Response: Species composition includes both species richness and abundance of each species. For the PCA, the reads were transformed into % abundance (reads of each species over total reads � 100%) for each species. The number of removed reads, merged reads, fungal reads and reads classified as other known taxa are provided in Supplementary Table 2. These information has been added to the text.

Comment 13: ‘The discussion section should include comparison with other studies on deep-sea hydrothermal vents as mentioned above, and a discussion on what environmental variables may have played a role in determining the fungal diversity. The current section “Putative ecological roles” is a misnomer because the majority of this section deals with either diversity alone or dispersal mechanisms. The section on Malassezia is really about its biogeography and not about its ecological roles. Therefore, I suggest restructuring the section into biogeography and dispersal mechanisms.’

Response: The section title has been changed from ‘Putative ecological roles’ to ‘Ecology of fungi’. A discussion between the results of this study and others from the deep-sea has been added. Jones (2000) has reviewed the key factors (temperature, salinity, etc.) influencing diversity of marine fungi but concerned only those occurring on organic substrates (mainly wood) in coastal habitats.

Comment 14: ‘“taxa” (line 33) is a vague term and should be avoided when possible. In the manuscript, taxa refer to species/genus in most occasions, and should be revised accordingly.’

Response: The term ‘taxa’ has been changed to ‘species’ in most cases, but is kept where necessary. 

Comment 15: ‘Line 64-65: it is misleading to state “especially in the deep sea”. I suggest removing this part.’

Response: This sentence has been modified and it reads ‘Yet, marine fungi are not represented in ocean ecosystem models [7], despite growing evidence of diverse marine fungi in the ocean.’

Comment 16: ‘Line 135: (The section on Collection of Samples). Table 1 should include information about which site and which type of substrate was sampled for cultivation.’

Response: These information is now available in Table 1.

Comment 17: ‘Line 220: “paired” should be “merged”.’

Response: This has been corrected.

Comment 18: ‘Line 256: “taxa” should be “species”.’

Response: This and others have been corrected.

Comment 19: ‘Table 2: what does the asterisk sign mean? It needs to be explained in the caption.’

Response: This is explained in the caption.

Comment 20: ‘Figure 3: it needs to be specified which sub-figures are for cultivation results and which sub-figures are for metabarcoding results.’

Response: These have already been specified in the y-axis and explained in the caption. No change is required.

Reviewer 2

Comment 21: ‘Please don't use the term culturomics! I would use culture-based through out the manuscript instead.’

Response: This term has been changed throughout.

Comment 22: ‘Additional coorections are annotated on the attached manuscript.’

Response: These have been corrected.

Comment 23: ‘Please include a summary or conclusion section at the end if possible.’

Response: A conclusion section is added and it reads ‘Fungal diversity in shallow hydrothermal vent system is a gap of our knowledge of the marine mycota. In this study, 54 and 49 species of the Ascomycota and the Basidiomycota were found in the black and yellow sediments collected at/near the shallow hydrothermal vent area of Kueishan Island by culture-based and metabarcoding methods, suggesting a high diversity of fungi in this extreme environment.’

---

## [Decision Letter · Decision Letter 1]

21 Nov 2019

PONE-D-19-20402R1

Insights into fungal diversity of a shallow-water hydrothermal vent field at Kueishan Island, Taiwan by culture-based and metabarcoding analysis

PLOS ONE

Dear Dr. Pang,

Thank you for submitting your manuscript to PLOS ONE. The authors have done a good job in addressing the majority of the comments from the reviewers and editor. I appreciate the effort that has been put into this. However, there are a few minor issues raised by both reviewers that I believe require clarification/revision prior to acceptance for publication. Therefore, we invite you to submit a revised version of the manuscript that addresses the points raised during the review process.

We would appreciate receiving your revised manuscript by Jan 05 2020 11:59PM. To enhance the reproducibility of your results, we recommend that if applicable you deposit your laboratory protocols in protocols.io, where a protocol can be assigned its own identifier (DOI) such that it can be cited independently in the future. For instructions see: http://journals.plos.org/plosone/s/submission-guidelines#loc-laboratory-protocols

We look forward to receiving your revised manuscript.

Kind regards,

Kin Ming Tsui

Academic Editor

PLOS ONE

Additional Editor Comments (if provided):

What would happen when sample 16-b (in PCA1) was excluded from the analysis? Have you done the comparison?

Reviewers' comments:

Reviewer's Responses to Questions

**Comments to the Author**

1. If the authors have adequately addressed your comments raised in a previous round of review and you feel that this manuscript is now acceptable for publication, you may indicate that here to bypass the “Comments to the Author” section, enter your conflict of interest statement in the “Confidential to Editor” section, and submit your "Accept" recommendation.

Reviewer #2: All comments have been addressed

Reviewer #3: (No Response)

2. Is the manuscript technically sound, and do the data support the conclusions?

Reviewer #2: Yes

Reviewer #3: Partly

3. Has the statistical analysis been performed appropriately and rigorously? 

Reviewer #2: Yes

Reviewer #3: Yes

4. Have the authors made all data underlying the findings in their manuscript fully available?

Reviewer #2: Yes

Reviewer #3: Yes

5. Is the manuscript presented in an intelligible fashion and written in standard English?

Reviewer #2: Yes

Reviewer #3: Yes

6. Review Comments to the Author

Reviewer #2: In the abstract:

1. The metabarcoding analysis amplifying a small fragment of the rDNA (from 18S to 5.8S) recovered 7-27 species from the black sediment and 12-27 species from the yellow sediment samples of the Ascomycota and the Basidiomycota.

Add respectively at the end with a comma.

2. In the abstract: One or two sentences to put results in a general context and broader perspective are missing.

3. Line 284 One common species were should be "one common species was isolated.

4. Figure legends are still not explaining the key results. Doing so will make the paper more interesting to the reader.

5. Conclusion statement is weak, please strengthen it.

Reviewer #3: (No Response)

7. PLOS authors have the option to publish the peer review history of their article (what does this mean?). If published, this will include your full peer review and any attached files.

Reviewer #2: No

Reviewer #3: No

---

## [Author Response · Author response to Decision Letter 1]

26 Nov 2019

Editor’s Comments:

Comment: ‘What would happen when sample 16-b (in PCA1) was excluded from the analysis? Have you done the comparison?’

Response: We have actually done the analysis without B16-b (please check the attached response to reviewers file as the figure cannot be shown here) and both axes were found to contribute significant variations in the sample B16-a. These suggest the samples collected at Site E were with very different species composition. We have modified the text as ‘PC1 axis contributed significantly to one of the two black sediment samples at site E (B16-b). If the outliner B16-b was removed from the analysis, both PC1 and PC2 contributed significantly to the other black sediment sample at site E (B16-a) (results not shown).’

Reviewer 1:

Comment 1: ‘However, there is a methodological issue which needs to be highlighted in the paper. Ideally the crab and egg samples would have been surface-sterilised before being ground up for culturing and molecular analysis. The results are still interesting and worth reporting, but it is important to discuss the fact that the cultures and DNA from these samples may simply be from spores resting on the surface of the crabs.’

Response: We agree with the reviewer. A sentence has been added in the discussion and it reads ‘It is also possible that the cultures isolated from the crabs (and the animal eggs) represented resting spores on the crab surface at the time of collection.’

Comment 2: ‘The authors make a very valid point in the discussion about airborne spores being found when using the metabarcoding methods. This is a really important point which is often missed in metabarcoding studies. It would be good to mention this point in the abstract.’

Response: A sentence has been added in the abstract and it reads ‘While some fungi found in this study were terrestrial species and their airborne spores might have been deposited into the marine sediment,…………’

Comment 3: ‘Introduction First sentence is too long. Please break it up.’

Response: The first sentence has been broken up into two sentences.

Comment 4: ‘Please define "deep sea".’

Response: The ‘deep-sea’ was loosely defined as ‘habitats below the epipelagic zone, (Herring 2001) and this has been added to the text.

Comment 5: ‘Please define "deep subsurface". How far below surface?’

Response: The deep subsurface habitats include both sedimentary and oceanic crustal habitats and this has been added to the text.

Comment 6: ‘Lines 59 to 60: Change " fungal species appears associated to the marine environment with 1255 species recently documented from the ocean [5]." to "fungal species appear to be associated with the marine environment with just 1255 species...".’

Response: This has been changed.

Comment 7: ‘Line 71: Change "... depending on the habitat, are thus facing many..." to "... depending on the habitat, thus face many...’

Response: This has been changed.

Comment 8: ‘Line 72: Change "...gradients, variable sea salt..." to "gradients and variable sea salt..."’

Response: This has been changed.

Comment 9: ‘Line 97: The word "illustrated" here doesn't seem right. Perhaps just use "found".’

Response: I cannot find the word ‘illustrated’ in the manuscript in Line 97. 

Comment 10: ‘Line 105: Change "...vents appear as unique..." to "...vents appear to be unique..."’

Response: This has been changed.

Comment 11: ‘Line 115: Change "28-29], however, knowledge..." to "28-29]. However, knowledge...".’

Response: This has been changed.

Comment 12: ‘Materials and methods: Ideally the crabs and eggs would have been surface sterilised before culturing and molecular analysis.’

Response: See Comment 1. 

Comment 13: ‘Line 143: Were the universal bottles sterilised?’

Response: Yes, the universal bottles were sterile. The word ‘sterile’ is added before universal bottles in the text.

Comment 14: ‘Line 146-147: How were the samples divided? Was there any randomisation process? If so, then please add this.’

Response: No randomisation process was performed. The text has been modified to clarify how the samples were divided and it reads ‘For each of the twenty-two sediment subsamples (in universal bottles) collected on the five collection dates, half of the sediment was transferred aseptically to another sterile universal bottle and freeze-dried for DNA extraction while the other half was used for isolation.’

Comment 15: ‘Table 1: Ideally all samples that are to be compared would be collected on the same day.’

Response: We agree with the reviewer on this. In fact, we have analysed twenty-two sediment subsamples in the metabarcoding analysis, but positive results were only obtained from 13 subsamples collected on the five collection dates. This is mentioned in the discussion.

Comment 16: ‘Line 174: Why was 25 deg C chosen? How does this compare to the temperature where the samples were taken?’

Response: From January 2014 to December 2017, the average water temperatures at Kueishan Island were between 20.2 °C and 29.4 °C. The use of 25 °C to incubate the inoculated samples was based on roughly the median of the lowest and highest average water temperatures.

Comment 17: ‘Line 175: Change "appeared" to "appearing".’

Response: This has been changed.

Comment 18: ‘Line 184: Change "scrapped" to "scraped".’

Response: ‘Scraped’ was used in the original text.

Comment 19: ‘Line 203: Change "Maxi Kit" to "Maxi Kits".’

Response: This has been changed.

Comment 20: ‘Results: Line 285: Change "were" to "was"’

Response: This has been changed.

Comment 21: ‘Line 291: Change "subjected to the DNA..." to "subjected to DNA..."’

Response: This has been changed.

Comment 22: ‘Line 297: Change "operational taxonomic unit" to "operational taxonomic units".’

Response: This has been changed.

Comment 23: ‘Line 314: Change "based the" to "based on the".’

Response: ‘based on the’ was used in the original text.

Comment 24: ‘Discussions: Line 395: Change " associated to" to "associated with".’

Response: ‘associated with’ was used in the original text.

Comment 25: ‘Line 440: Change "Fungal community" to "Fungal communities".’

Response: This has been changed.

Comment 26: ‘Line 443: Change "habor" to "habors".’

Response: This has been changed.

Comment 27: ‘Line 463: Change "majority has been" to "majority have been".’

Response: This has been changed.

Comment 28: ‘Line 484 to 486: This is an interesting point. Please expand on this and provide references for the reader to follow up.’

Response: We have elaborated on the topic and the text reads as ‘Occurrence of filamentous basidiomycetes in this report and in other studies may suggest that spores of the terrestrial Polyporales (e.g. Bjerkandera adusta, Cerrena sp., Phanerochaete tuberculate, Phlebia chrysocreas, Rigidoporus sp., Trametes cubensis and T. versicolor) and Agaricales (e.g. Chondrostereum sp. and Schizophyllum commune) deposit in the marine sediment [19�20]. Kueishan Island is only ~10 km away from the main Taiwan Island and it is likely that spores of the filamentous basidiomycetes were originated from the terrestrial environment through freshwater runoff or air deposition [61] and deposited into the sediment.’

Reviewer 2: 

Comment 29: ‘In the abstract: 1. The metabarcoding analysis amplifying a small fragment of the rDNA (from 18S to 5.8S) recovered 7-27 species from the black sediment and 12-27 species from the yellow sediment samples of the Ascomycota and the Basidiomycota. Add respectively at the end with a comma.’

Response: The meaning of the sentence is not clear as written. This has been rewritten as ‘The metabarcoding analysis amplifying a small fragment of the rDNA (from 18S to 5.8S) recovered 7-27 species from the black sediment and 12-27 species from the yellow sediment samples and all species belonged to the Ascomycota and the Basidiomycota.’

Comment 30: ‘2. In the abstract: One or two sentences to put results in a general context and broader perspective are missing.’

Response: An extra sentence has been added and it reads ‘This study is the first to observe a high diversity of fungi associated various substrates at a marine shallow water hydrothermal vent ecosystem.’

Comment 31: ‘3. Line 284 One common species were should be "one common species was isolated.’

Response: This has been changed. 

Comment 32: ‘4. Figure legends are still not explaining the key results. Doing so will make the paper more interesting to the reader.’

Response: The figure legends have been elaborated.

Comment 33: ‘5. Conclusion statement is weak, please strengthen it.’

Response: The conclusion has been rewritten.

---

## [Editor Report · Decision Letter 2]

4 Dec 2019

Insights into fungal diversity of a shallow-water hydrothermal vent field at Kueishan Island, Taiwan by culture-based and metabarcoding analysis

PONE-D-19-20402R2

Dear Dr. Pang,

We are pleased to inform you that your manuscript has been judged scientifically suitable for publication and will be formally accepted for publication once it complies with all outstanding technical requirements.

With kind regards,

Kin Ming Tsui

Academic Editor

PLOS ONE

Additional Editor Comments (optional):

Thank you for the revision. The revised manuscript could be accepted for publication.
---

## [Editor Report · Acceptance letter]

10 Dec 2019

PONE-D-19-20402R2 

Insights into fungal diversity of a shallow-water hydrothermal vent field at Kueishan Island, Taiwan by culture-based and metabarcoding analyses 

Dear Dr. Pang:

I am pleased to inform you that your manuscript has been deemed suitable for publication in PLOS ONE. Congratulations! Your manuscript is now with our production department. 

With kind regards,

on behalf of

Dr. Kin Ming Tsui 

Academic Editor

PLOS ONE